



# Modelling the effects of climate and landcover change on the hydrologic regime of a snowmelt-dominated montane catchment

Russell S. Smith[1], Caren C. Dymond[2], David L. Spittlehouse[3], Rita D. Winkler[3], and Georg Jost[4]

[1]WaterSmith Research Inc., Kelowna, V1Y 5N3, Canada
[2]Ministry of Forests, Victoria, V8W 9C5, Canada
[3]Retired, Ministry of Forests, Victoria, V8W 9C5, Canada
[4]Generation System Operations, BC Hydro, Burnaby, V3N 4X8, Canada

*Correspondence to*: Russell S. Smith (rsmith@watersmith.ca)

**Abstract.** Climate change poses risks to society through the potential to alter peak flows, low flows, and annual runoff yield.
Wildfires are projected to increase due to climate change; however, little is known about their combined effects on hydrology. This study models the combined impacts of a climate change scenario and multiple landcover scenarios on the hydrologic regime of a snowmelt-dominated montane catchment, to identify management strategies that mitigate negative impacts from climate and/or landcover change. The combination of climate change and stand replacing landcover disturbance in the middle and high elevations is predicted to advance the timing of the peak flow three to five times more than the advance generated by
disturbance alone. The modelling predicts that the combined impacts of climate change and landcover disturbance on peak flow magnitude are generally offsetting for events with return periods less than 5-25 years, but additive for more extreme events. There is a dependency of extreme peak flows on the distribution of landcover. The modelling predicts an increasing importance of rainfall in controlling peak flow response under a changing climate, at the expense of snowmelt influence. Extreme summer low flows are predicted to become commonplace in the future, with most of the change in frequency
occurring by the 2050s. Low annual yield is predicted to become more prevalent by the 2050s, but then fully recover or become less prevalent (compared to the current climate) by the 2080s, because of increased precipitation in the fall-spring period. The modelling suggests that landcover disturbance can have a mitigative influence on low water supply. The mitigative influence is predicted to be sustained under a changing climate for annual water yield, but not for low flow. The study results demonstrate the importance of a holistic approach to modelling the hydrological regime rather than focusing on a particular component.
Moreover, for managing watershed risk, the results indicate there is a need to carefully evaluate the interplay among environmental variables, the landscape, and the values at risk. Strategies to reduce one risk may increase others, or effective strategies may become less effective in the future.



# 1       Introduction

Climate change poses risks to society due to potential increases in flooding and reduced water supply (Burn et al., 2016; Gaur et al., 2019). Concurrently, wildfires are projected to increase in severity and extent in many parts of the world due to climate change, with potentially devastating hydrologic and geomorphic consequences (Flannigan et al., 2009; Wang et al., 2015). Few studies have addressed the combined impacts of climate and landcover changes on hydrologic response, with even fewer also addressing multiple aspects of the runoff regime, including peak flows, low flows, and overall runoff yield. Doing so

would facilitate a holistic evaluation of potential long-term changes in hydrology and threats to communities, infrastructure, and aquatic habitat, as well as the potential to identify adaptation strategies.

Climate change in snowmelt-dominated mountainous regions is likely to result in decreased peak snowpack depths, earlier spring snowmelt, more frequent winter rain events, and more persistent midwinter melt, resulting in more transient snowpacks

and higher rates of winter runoff (Elsner et al., 2010; Merritt et al., 2006; Vano et al., 2010). Shallower snowpacks and earlier melt often result in lower peak flows, a shift towards less frequent flooding, and decreased summer low flows and overall yield (Burn et al., 2010; Dierauer et al., 2018; Merritt et al., 2006). These studies illuminate potential hydrologic impacts related to large scale (spatial and temporal) changes in air temperature and precipitation; however, few studies have accounted for potential increases in the intensity of weather patterns at synoptic or event scales. Westra et al. (2013) showed that nearly two-

thirds of rainfall stations globally exhibited increasing trends in annual maximum daily precipitation between 1900 and 2009, with corresponding increases in air temperature. The higher latitudes showed stronger positive associations, with values ranging between approximately 7.5% and 13% K-1 in the Northern Hemisphere above 50ºN. Donat et al. (2016) had similar findings, with climate projections for the rest of the century showing continued intensification, particularly for dry regions. We suggest that more intense rainfall during periods of high catchment wetness (e.g., ripe snowpacks and/or wet soils) could

increase flood frequencies and overall runoff yield.

Flooding and water supply are also affected by landcover change resulting from wildfire, as well as insects, disease, and forest harvesting (Robinne et al., 2021; Saxe et al., 2019; Schnorbus and Alila, 2013). Forest cover disturbance typically increases snowpack accumulation and ablation rates, advances the timing of melt, and increases runoff yield (Winkler et al., 2017;

Winkler et al., 2015). Catchments with more than 20% of the forest cover disturbed have shown peak flow increases; however, the magnitude of response is highly variable, illuminating nonlinear response behaviour (Adams et al., 2012; Goeking and Tarboton, 2020; Schnorbus and Alila, 2013). For instance, several studies have shown that the difference in peak flows between disturbed and undisturbed catchments decreases with increasing event magnitude (Bathurst et al., 2011a; Bathurst et al., 2011b; Moore and Wondzell, 2005), whereas some have shown the difference to increase with increasing event magnitude (Moore

and Wondzell, 2005; Schnorbus and Alila, 2013). Yet others have shown a decrease in yield caused by disturbance (Adams et al., 2012; Goeking and Tarboton, 2020). Decreasing yield has typically been associated with non-stand replacing disturbance



and/or arid climates. Low flows have often been shown to increase immediately post-disturbance, followed by longer-term decreases as evapotranspiration (ET) increases with forest regeneration, though the changes are often not significant (Coble et al., 2020; Goeking and Tarboton, 2020). These findings indicate that the impacts of landcover disturbance on runoff yield are
complex, with variability in the direction and magnitude of change dependent on the specific landcover and hydrometeorological conditions considered.

With respect to the combined influences of climate and landcover, a global study of large watersheds by Li et al. (2017) showed that climate and landcover played equal roles in annual yield variations. Among 67 watersheds, 51 exhibited additive effects
of climate and landcover change, and 16 showed offsetting effects, with the former generating a higher risk of extreme outcomes (floods or droughts). They also found that smaller and dryer watersheds are hydrologically more sensitive to landcover change than larger and wetter watersheds. Among forest-dominated regions of the world, Wei et al. (2018) found that the global mean variation in annual runoff due to landcover change was 30.7%, whereas 69.3% was attributed to climate change. For British Columbia, Canada, large scale mountain pine beetle infestation and salvage logging accounted for 39.0%
of the variation in annual runoff, compared to 61.0% for climate change.

The purpose of the current study was to assess the combined impacts of climate and landcover changes on the hydrologic regime of the greater Penticton Creek Watershed in southwest Canada, and to identify management strategies that mitigate negative impacts from either change. The catchment is located in a semi-arid, mountainous region. Vegetation ranges from
grassland in the lower elevations, to dense coniferous forest at higher elevations. Wildfires are frequent (BC Government, 2019) and fire weather conditions are expected to become more severe as the climate warms (Nitschke and Innes, 2008; Spittlehouse and Dymond, 2022). Climate envelope modelling indicates a potential for reduced canopy density in the lower elevations (Wang et al., 2012). Both flood and water supply risks are a concern, and climate and landcover changes are concerns for impacting these risks. Meanwhile, Penticton Creek provides water to a town of 34,000 people that has a bustling tourism
industry, and irrigation to wineries and fruit orchards.

To achieve the study objectives, we (1) developed a catchment runoff model using the Raven hydrological modelling framework (Raven) (Craig et al., 2020); (2) combined three different climate conditions (i.e., one emission pathway with three different time periods) and five different landcover conditions to assess combined impacts on snowpack dynamics, runoff
timing, and the frequency of peak flows, low flows, and annual runoff yield; and (3) evaluated implications to flooding and water supply limitations, upland water storage, and watershed management. Each landcover condition was treated as static (i.e., no change over time) and, thus, represents a specific example of landcover disturbance.



## 2    Methods

### 2.1    Study catchment

The study catchment for hydrologic modelling comprised the area draining to the Penticton Creek below Harris Creek streamflow gauging station (Water Survey of Canada [WSC] Stn. 08NM170, 148 km2, https://wateroffice.ec.gc.ca/search/ historical_e.html) (Fig. 1). Elevations in the catchment range from approximately 593 m to 2154 m, and drain from the Okanagan Plateau, an area of relatively subdued mountainous topography. Stream discharge data were also used for three headwater sub-catchments, including 240 (08NM240) and 241 Creeks (08NM241) (both southerly facing), and Dennis Creek

(08NM242) (westerly facing); and for Greyback Lake (controlled reservoir) (08NM169) (hereafter, all discharge values refer to the main catchment outlet, unless specified otherwise).

The underlying bedrock is comprised primarily of intrusive rock, with areas of metamorphic rock in the north and southwest. The surface geology is dominated by glacial till. Soils fall into three main types, including sandy loam over bedrock at ~0.65

m, deep loam to ~1.2 m, and sandy loam over sand at ~0.6 m (Fig. 2) (https://sis.agr.gc.ca/cansis/index.html). A deeply incised mainstem channel runs northeast-southwest, with tributary streams forming a dendritic drainage pattern.





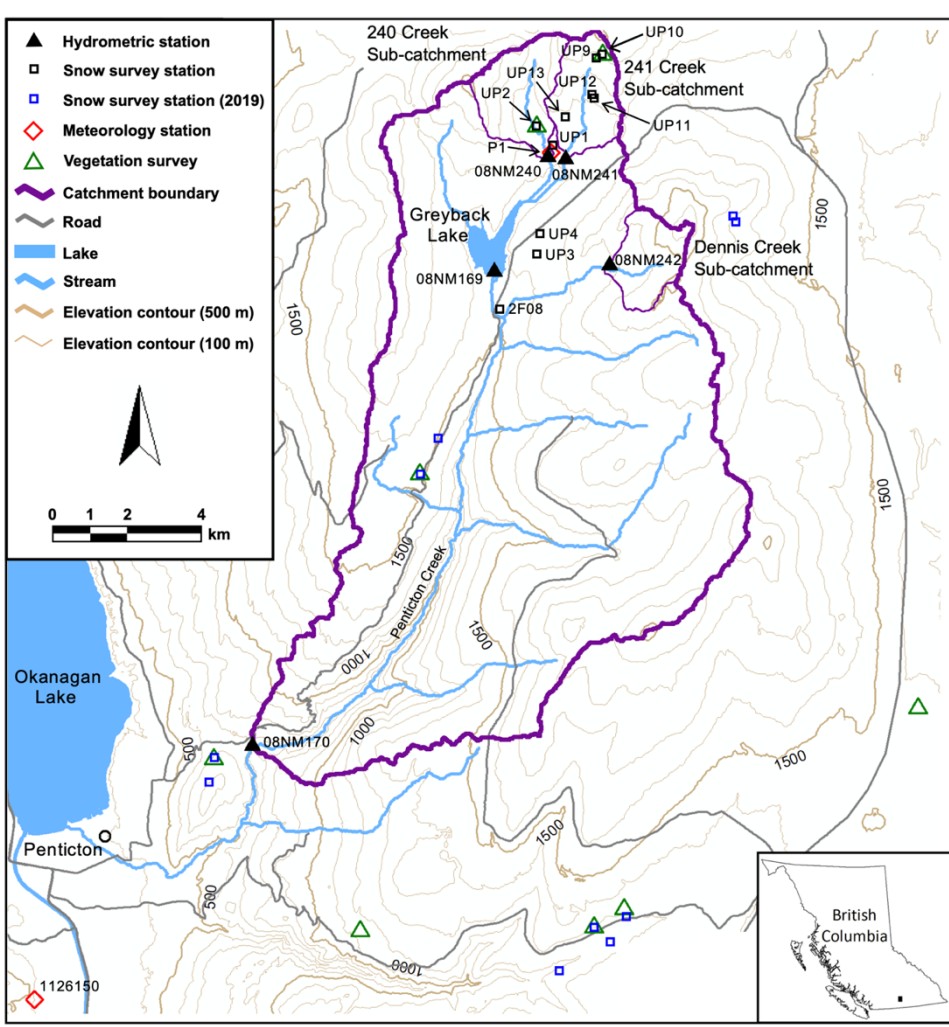

**Figure 1: Location and monitoring sites of the Penticton Creek study catchment. Snow survey station (2019) refers to data acquired by Smith (2022) (Table S2).**



**Figure 2: Physiography of the study catchment. Tree species (panel 'c') and crown closure (panel 'd') represent the estimated pre-disturbance conditions. Harvest year (panel 'f') and fire year (panel 'g') represent the actual disturbance histories.**


Five Biogeoclimatic Ecosystem Classification (BEC) zones are represented in the study catchment including Ponderosa Pine (PP), Interior Douglas-fir (IDF), Montane Spruce (MS), Engelmann Spruce – Subalpine Fir (ESSF), and Interior Mountain-heather Alpine (IMA) (Fig. 2) (Lloyd et al., 1990). The IDF and ESSF include two and three variants, respectively. Dominant tree species in the study catchment include ponderosa pine (Pinus ponderosa) (Py), Douglas-fir (Pseudotsuga menziesii) (Fd),

lodgepole pine (Pinus contorta) (Pl), hybrid white spruce (Picea glauca x engelmannii) (Sxw), Engelmann spruce (Picea engelmannii) (Se), and subalpine fir (Abies lasiocarpa) (Bl).



The catchment has a history of forest cover disturbance (Fig. 2). Between 1970 and 1976, approximately 3.7% of the catchment
was harvested, followed by 0.2% in 1984, 0.2% in 1991, and 10.4% between 1992 and 2012. Four wildfires occurred between
1919 and 1931 (covering 17% of the drainage), followed by fires in 1970 (30%) and 1994 (2%). The 1970 fire was located in
the arid and sparsely forested lower elevations.

A snowpack persists at the middle and upper elevations from October-November through April-June, and is intermittent at the
lowest elevations. This climatic range is reflected in the air temperature and precipitation (Table 1). Spring snowmelt dominates
the hydrologic regime, but intense rainfall events during spring freshet can also contribute substantially to peak flows. Mean
annual runoff averages 265 mm.

**Table 1: Mean annual air temperature (°C) and total annual precipitation (mm) for the current climate at the P1 and Penticton**
**Airport meteorological stations, and the incremental change projected for the future climates at the P1 station (adapted from**
**Spittlehouse and Dymond, 2022; all values derived from 1983-2016 meteorological records).**

| Station (climate condition) | Mean air temperature (°C) | | | | Precipitation (mm) | | | |
|---|---|---|---|---|---|---|---|---|
| | Winter | Spring | Summer | Fall | Winter | Spring | Summer | Fall |
| P1 (Current) | -5.3 | 3.5 | 11.2 | -1.3 | 186 | 240 | 161 | 205 |
| Penticton Airport (Current) | 0.8 | 12.1 | 19.7 | 5.6 | 78 | 113 | 86 | 86 |
| P1 (2050s) | +2.4 | +2.1 | +4.0 | +3.1 | +0 | +8 | -57 | +32 |
| P1 (2080s) | +5.1 | +4.3 | +6.7 | +4.9 | +31 | +17 | -63 | +67 |

## 2.2 Representing the catchment

### 2.2.1 Raven setup

Raven was set up as a process-oriented deterministic runoff model, and run at a daily time-step. The study catchment was
spatially discretized using a semi-distributed approach. Precipitation and air temperature were distributed from a single
meteorological station, and accounted for orography and rain-snow partitioning. The snowpack balance incorporated coupled
mass and energy balance equations. The full snowpack energy balance was represented using algorithms that estimate energy
fluxes using daily precipitation, and daily minimum and maximum air temperature. It accounted for cloud cover, short-wave
radiation, long-wave radiation, and turbulent flux.

Vegetation was represented as a single canopy layer, and accounted for canopy interception, canopy drip,
sublimation/evaporation of intercepted precipitation, and transpiration (Penman Monteith). In-catchment runoff was routed to
the sub-catchment outlet assuming a two-layer soil. It accounted conceptually for individual controls on runoff generation,



including soil evaporation, overland flow, interflow, percolation, and baseflow. Surface runoff was subsequently routed to the main catchment outlet by in-channel flow. The historical record of streamflow at the catchment outlet was adjusted for ongoing storage changes in Greyback Lake, allowing the catchment to be represented as an unregulated system. The intent of this adjustment was to isolate the effects of climate and landcover change on the runoff regime, without the confounding effects of reservoir operations.

### 2.2.2 Meteorology

#### 2.2.2.1 Observed record

The historical record from the P1 weather station (1619 m) in the Upper Penticton Creek Watershed Experiment (UPC) (Winkler et al., 2017) (Fig. 1) was used for model parameterization. The P1 station has a daily record spanning 1992-2016, but was extended back to 1970 using other UPC stations (1983-1991 period) and regional records (1970-1983) (Winkler et al.,

2017). The 1970-2014 portion of the extended record was used in this study to coincide with available hydrometric and snowpack records.

#### 2.2.2.2 Climate change scenario

The climate change scenario was obtained from the Bias Corrected Constructed Analogues with Quantile mapping (BCCAQ-

v1) data sets (PCIC, 2014; Sobie and Murdock, 2017) for a grid cell that includes the location of the P1 weather station. These data are CMIP5 general circulation model output (Taylor et al., 2012) of daily precipitation, and daily minimum and maximum air temperature, downscaled to 300 arc seconds (approximately 10 km). The scenario (CSIRO-Mk3-6-0-r1 RCP8.5) (CSIRO85) is in the upper 90th percentile of the range of projections for warmer and drier summers, and increased fire risk in the southern Okanagan (Spittlehouse and Dymond, 2022). It was chosen to investigate how a large change in climate could

impact the hydrology of Penticton Creek Watershed.

The climate conditions represent no change (current climate), moderate change (2041-70 period; 2050s), and severe change (2071-2100 period; 2080s) in weather and fire risk. The mean annual precipitation in the 2050s and 2080s is projected to decrease by 2.0% and increase by 6.7% from current conditions, and the mean annual air temperature is projected to increase

by 2.9°C and 5.3°C, respectively. The scenario projects substantial seasonal changes in precipitation, including drier summer in the 2050s and 2080s; wetter fall in the 2050s; and wetter fall, winter, and spring in the 2080s (Table 1).

#### 2.2.2.3 Synthetic record

100-year weather records for a stationary climate (current, 2050s, and 2080s) were required for long-term hydrologic

simulations and subsequent estimation of return periods for extreme events. Synthetic weather records for current and projected future conditions were created using the LARS-WG5 weather generator (Semenov and Barrow, 2013; Semenov and Stratonovitch, 2010). For the current climate, it was calibrated with 32 years (1984-2016) of daily minimum and maximum air



temperature and precipitation data for the P1 station. Compared to the historical observations, the simulated values tended to underestimate the statistical distribution of the coolest temperatures (Spittlehouse and Dymond, 2022). Consequently, a

quantile mapping procedure was used to adjust the synthetic records (Sobie and Murdock, 2017). The procedure involved dividing the synthetic and measured temperature datasets into 5% quantiles, and regressing against each other to generate seasonally based non-linear adjustment equations. In contrast to temperature, the distribution of daily precipitation and monthly 1-day extreme precipitation, and the length of maximum monthly dry and wet periods were simulated well, and no quantile adjustment was required.


For the 2050s and 2080s climates, the ratio of future to base precipitation, and wet and dry period length were used in LARS-WG5 to adjust the statistical distributions of these variables. The resulting time series were assessed to ensure the mean monthly change in temperature and precipitation matched that projected by the climate model. More information on the synthetic record is provided by Spittlehouse and Dymond (2022).

### 2.2.3 Landcover


The landcover conditions implemented in Raven included the historical disturbed forest cover conditions from 1976 and 2012, an estimate of the actual pre-disturbance condition (hereafter referred to as the forested condition), and the forested condition combined with two simulated wildfire conditions (hereafter referred to as the small burn and large burn conditions) (Fig. 3, Table 2). The forested condition was considered the baseline condition for modelling purposes. It was reconstructed from 2012

Vegetation Resources Inventory (VRI) forest cover data (BC Government, 2012) by interpolating between mature stands after removing mapped disturbance areas.



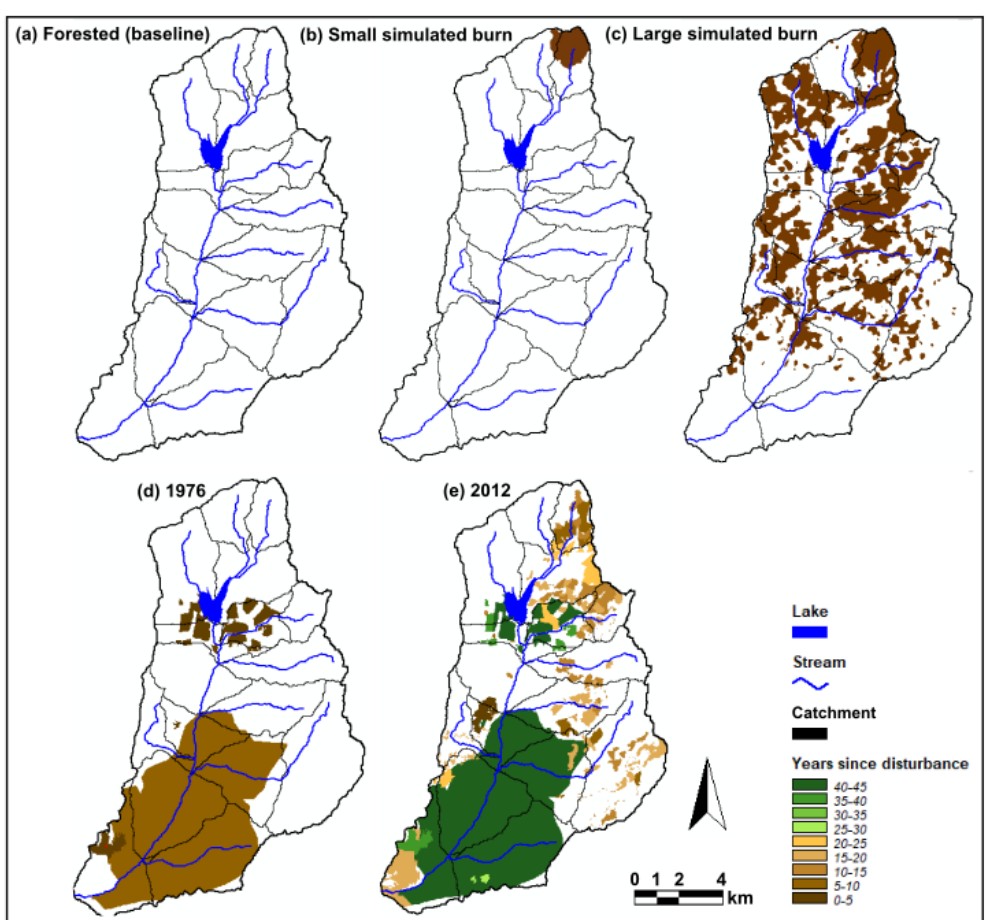

**Figure 3: Spatial representation of the historical and simulated disturbance histories implemented in Raven. Panels 'd' and 'e' represent the disturbance histories represented in the years 1976 and 2012, respectively.**



**Table 2: Summary of disturbance area and LAI by elevation for the landcover conditions used in the long-term simulations.**

| Landcover condition | [1]Disturbed area (%) | [2]Leaf area index by elevation (m) | | | | |
|---|---|---|---|---|---|---|
| | | <1300 | 1300-1650 | 1650-1900 | >1900 | All elev. |
| Forested | [3]0 | 0.72 | 1.77 | 1.91 | 1.62 | 1.69 |
| Small burn | [4]2 | 0.72 | 1.77 | 1.86 | 1.52 | 1.66 |
| Large burn | [4]32 | 0.65 | 1.24 | 1.12 | 1.19 | 1.11 |
| 1976 | [5]34 | 0.35 | 1.21 | 1.86 | 1.62 | 1.41 |
| 2012 | [6]47 | 0.83 | 1.70 | 1.65 | 1.48 | 1.55 |

1. Portion of catchment impacted by wildfire or forest harvesting
2. HRU area weighted mean of parameter values. The percentage of watershed area comprised by each elevation range is 13, 37, 43, 7, and 100 respectively.
3. Areas burned between 1919 and 1931 were excluded because of stand regeneration.
4. Incorporates only the burned areas that were simulated as stand replacing (Sect. 2.2.3).
5. The 1970 burn occurred in areas with pre-existing stands that were naturally sparse.
6. Includes areas disturbed in the 1970s and 1980s that experienced substantial stand regeneration by 2012.

The 1976 and 2012 conditions resulted from a combination of historical wildfires and forest harvesting, with the 2012 condition incorporating several years of forest regeneration (i.e., hydrologic recovery) for older disturbance areas (Fig. 3). Some regenerating stands had higher densities than the neighbouring mature stands, resulting in the 2012 condition having higher stand densities at lower elevations than the forested condition (Table 2).

The burned areas for the small and large burn conditions were simulated using LANDIS-II, Forest Carbon Succession extension (Dymond et al., 2016), and the Dynamic Fuels Fire system (Sturtevant et al., 2009). Initial communities were based on 2014 VRI data (BC Government, 2014a). Input variables of establishment, net primary productivity, and maximum biomass were generated by Tree and Climate Assessment models (Dymond et al., 2016; Nitschke and Innes, 2008). Fire regime parameters were derived for each BEC variant (upper three variants were amalgamated due to small size) using historical fire data (points and polygons) from 1950 to 2013 over a 2.25 million ha management unit, which includes the Penticton Creek Watershed (BC Government, 2014b). A 100 m digital elevation model (DEM) was incorporated into the fire spread modelling (BC Government, 2014c). LANDIS-II was run repeatedly, taking advantage of the stochastic nature of the simulations until two scenarios occurred with a small and large burn (relative to the size of the study catchment only, as larger fires occurred in the management unit) (Fig. 3). For representing landcover patterns in Raven, only the most severe fire intensities, which caused total canopy mortality, were considered disturbed. It is acknowledged that moderate burn severities can result in substantial mortality; however, this approach was followed to limit the quantity and complexity of the landcover scenarios.



### 2.2.4    Spatial discretization

A total of 1,315 hydrological response units (HRUs) were discretized for representing the catchment (Fig. S1). Each of the HRUs was characterized by a unique set of parameter values representing area, elevation, vegetation, soil, slope gradient, and slope aspect. The size of the HRUs averaged 0.11 km2, with the largest HRU being 2.24 km2. Table S1 (Supplementary) summarizes the spatial data used to discretize the HRUs. The spatial data (25 m grid) were processed using a combination of SAGA GIS (Conrad et al., 2015) and R statistical software (version 3.5.0) (R Core Team, 2020).


In the process of discretizing the HRUs, a total of 17 drainage units were delineated to separate the study catchment into sub-catchments (Fig. 2). Additional boundaries were imprinted on the catchment discretization to distinguish BEC variant, soil type, and Greyback Lake (Fig. 2). Burned (historical and simulated) and harvested areas (Fig. 3) were also imprinted, as well as vegetation type (stratified by leading tree species and density). Finally, a 2000 m by 2000 m grid was imprinted on the

discretization to limit the maximum possible size of an HRU.

For imprinting BEC variant, the IMA and ESSFdcp were combined with the slightly lower elevation ESSFdcw variant (hereafter referred to as ESSFH), due to the small area encompassed by the upper two variants and the lack of available data for discriminating these variants in the model parameterization. However, these upper elevation BEC variants were

distinguished from the lowest elevation ESSF variant (ESSFdc1) (hereafter referred to as ESSFL), as the upper elevation stands have a more clumped distribution of trees. Clumpiness creates large canopy openings that can shade the snowpack and attenuate wind velocities, while offering reduced capacity to intercept precipitation. Field observations suggest that this stand condition at the upper elevations can result in net negative snowfall interception (comparing to snowpack accumulation in clearings). The IDF variants and the PP were also combined (hereafter referred to as IDFPP) for similar reasons as discussed

above for the ESSFH. The ESSFH and IDFPP was each stratified by leading tree species and density to account for important influences on hydrology.

### 2.3    Model parameterization

For each HRU, the mean elevation, slope gradient, canopy closure, and tree height, and the median slope aspect were calculated from spatial data (Table S1). Wherever possible, other model parameters were set based on empirical observations. Data

sources included field observations and modelling experience from the study catchment and other catchments in the southern interior of BC (Smith, 2018, 2022), information available in guidance documents (Craig and the Raven Development Team, 2022; Quick, 1995), and scientific literature (examples cited below). Parameters with substantial uncertainty and/or sensitivity, and those associated with conceptually oriented algorithms (e.g., runoff routing) were set through calibrating simultaneously on in-catchment snowpack water equivalent (SWE) and stream discharge, and precipitation at Penticton Airport (Fig. 1).



Parameter ranges were constrained as permitted by available data, including the use of regional SWE data from outside the study catchment (Fig. 1; Table S2) (Smith, 2022).

The parameterization was strengthened by the use of (1) SWE data encompassing many physiographic contrasts, including elevations ranging from 710 m to 1922 m, varying hillslope orientations, clearings and mature stands, and low through high
stand densities (Fig. 1; Table S2); and (2) a nested structure to the discharge data, incorporating variation in scale, catchment orientation, soil type, BEC variant, tree species, and stand density (Fig. 1). This multipronged approach provided a large amount of information for parameterizing a range of internal model processes.

### 2.3.1    Meteorology & energy balance

Parameters were calibrated that correspond to air temperature and precipitation gradients, attenuation of solar radiation,
atmospheric stability, snowpack albedo, and snowpack energy storage. Parameter constraints were informed by historical measurements of snowpack albedo at UPC (Spittlehouse and Winkler, 2004), and by empirical relations developed from historical hourly records of air temperature, precipitation, and solar radiation.

### 2.3.2    Vegetation

Some tree species were grouped for parameterizing mature stands, but were subsequently stratified by BEC variant and stand
density, which resulted in the groupings at some elevations being dominated by certain species. The groupings were formed because of a tendency for stands to have a mix of species, and because of limitations in available data for parameterizing vegetation (e.g., availability of forest-clearing paired SWE for different combinations of species, density, and elevation). Groupings included Sxw, Se, and Bl as one type (spruce-fir, S/B); and Py, Fd, and Pl as another type (pine-fir, P/F). At middle and high elevations, P/F was represented primarily by Pl. At low elevations, P/F was represented primarily by Py and Fd.


The density of mature stands was classified using canopy closure values from VRI forest cover data (Table S1) as follows: clearings (<20% canopy closure), low density (20-40%) (LD), moderate density (40-68%) (MD), and high density (68-95%) (HD). Regenerating stands were classified based strictly on tree height due to a lack of stand-specific density data for the regenerating areas. Combining all factors used to differentiate stand types (species, BEC variant, canopy closure for mature
stands, tree height for regenerating stands), the historical landcover conditions were represented using a total of 23 vegetation classes.

Parameter values for canopy closure and leaf area index (LAI) were informed by relating canopy closure values from the VRI forest cover data (Table S1) to canopy closure and LAI values obtained from hemispherical photos in the field. Seven plots
were established in mature forests representing the following BEC variants: ESSFdc, MSdm1, IDFdm1, IDFxh1, and PPxh1. LAI is an important influence on the snowpack energy balance (and, thus, the volume and timing of snowmelt and runoff)



through extinction of solar radiation. For these reasons, LAI values were further informed through a tightly constrained calibration on SWE and discharge. Throughfall was also calibrated on SWE and discharge for six vegetation classes that were most important for influencing the water balance. Throughfall parameters for the remaining vegetation classes were manually

assigned based on the calibrated throughfall values, accounting for differences in stand characteristics and elevation (i.e., throughfall percentage is expected to decrease with increasing precipitation). Wildfire disturbance and clearcut harvesting were treated the same (i.e., as clearings) with respect to impacts on LAI, throughfall, and canopy closure.

Field knowledge of local conditions and hydrometeorological processes was also considered for parameterizing vegetation.

For instance, constraints on LAI and throughfall parameters were adjusted for mature stands in the ESSFH to account for more clumpy stand structures, compared to the ESSFL (Sect. 2.2.4). Greater wind exposure at higher elevations was also considered for parameterizing canopy interception, due to influences on snowpack unloading from trees. Values published in literature were also considered (Brockley and Simpson, 2004; Kollenberg and O'Hara, 1999; Pugh and Gordon, 2012).

### 2.3.3  Calibration & validation

Various time periods were used for optimizing parameters on different data types. The 1971-1981 period was used for optimizing on discharge at the main catchment outlet (adjusted for storage in Greyback Lake). The 1984-1992 period was used for optimizing on sub-catchment discharge (240, 241, and Dennis Creeks). The 1995-1997 and 2009-2014 periods were used for optimizing on SWE, to incorporate data from clearings and a range of regenerating and mature stand types. The forest cover conditions in the catchment were relatively static over each period (Fig. 2). A split sample calibration-validation

approach was implemented. Years with high or low water yield, and cool or warm phases of the Pacific Decadal Oscillation (PDO) were divided more-or-less evenly between calibration and validation sets.

Parameter optimization was implemented through Ostrich using the Dynamically Dimensioned Search algorithm (Matott, 2005). A composite objective function was used for evaluating model performance (i.e., goodness of fit) using the following

metrics:
- ▪ Nash-Sutcliffe Efficiency (NSE) calculated for spring freshet discharge (March-July), and again after smoothing the data using a 15 day running mean;
- ▪ absolute bias in overall spring freshet yield;
- ▪ absolute bias in overall low flow yield (August-February);

- ▪ NSE and absolute bias for SWE; and
- ▪ absolute bias in annual precipitation at Penticton Airport.

Each metric was calculated for individual years, then averaged among years to minimize the potential for compensatory effects. Smoothed discharge data were used to emphasize the quality of the multi-day fit through the spring freshet. Absolute bias was used to ensure the model did not inflate or deflate overall yield to obtain higher NSE values. Precipitation bias was used to





constrain precipitation lapse rates for achieving a plausible representation of the water balance in the lower catchment area. Spring freshet and low flow discharge were evaluated separately to ensure reasonable predictive performance for both high and low flows. Model suitability was evaluated based on multiple factors, including the goodness of fit statistics, visual inspection of plotted output, and the plausibility of the parameter values in "physical space" (for physically oriented algorithms).

## 2.4  Long-term simulations

Each of the three synthetic meteorological datasets (i.e., climate conditions) (Table 1) was paired separately with each of the five landcover conditions (Fig. 3; Table 2) to generate 15 different modelling scenarios (recall that each landcover condition was treated as static during the simulations). Discharge and SWE were, thus, simulated over a 100 year period for each scenario. The first year of simulation was used as a warm-up (i.e., spin-up) to ensure the soils were suitably "wet" leading into

the subsequent simulation years. Accordingly, the first year of discharge output was discarded from all analyses.

## 2.5  Snowpack sensitivity analyses

A point-scale snowpack sensitivity analysis was implemented to examine the influences of vegetation and topography on snowpack dynamics, and changes under the future climate conditions. Site characteristics were selected to represent topographic end-points within the range of typical conditions existing in the study catchment. The selected characteristics

included south and north facing sites (i.e., high and low solar exposures) on a 40% slope gradient at elevations of 1100 m, 1500 m, and 1800 m (hereafter referred to as low elevation, middle elevation, and high elevation). Two vegetation types were simulated for each elevation, including a clearing, and the most prevalent mature forest type at the specific combination of elevation and solar exposure. Results were analysed for annual maximum SWE and timing of snowpack melt-out (first day of the year with less than 25 mm of SWE).


A catchment scale snowpack sensitivity analysis was also implemented to further examine the influences of vegetation, topography, and climate on snowpack dynamics. To examine the climate effect, annual maximum SWE and the timing of snowpack melt-out were mapped catchment-wide for the forested condition under the current, 2050s, and 2080s climates. To examine changes in the disturbance effect under a changing climate, the difference in SWE and melt-out between 100% cleared

and forested catchment conditions were mapped for each climate.

## 2.6  Frequency analysis

Intensity-duration-frequency analysis was implemented on the synthetic precipitation records. Event frequency analysis (EFA) was implemented on output from the long-term simulations for peak flow, summer low flow (based on 30-day mean discharge), and annual discharge (arithmetic mean of the discharge over a given water year). For peak flow, higher runoff was treated as

more extreme due to risks to social (e.g., residential and commercial structures, stream crossings) and environmental (e.g.,


integrity of fish habitat) values. For low flow and annual discharge, lower runoff was treated as more extreme due to risks to water supply (e.g., domestic consumption, ecological flows).

The intensity-duration-frequency analysis and the EFA involved statistically fitting frequency curves to annual data series based on the Log Pearson Type III statistical distribution (using the R statistical programming language; (Craig and the Raven Development Team, 2022). The Gringorten extreme value plotting position formula was used for plotting annual data (Gringorten, 1963).

# 3 Results

## 3.1 Model performance

The overall fit to the observed data was generally good for both the calibration and validation periods, based on visual inspection of the plots (Fig. S2 and S3; Supplementary) and the performance statistics (Table S3). The composite objective function was 0.87 for the calibration and 0.79 for the validation. The overall level and timing of spring freshet runoff was matched well in most years at the catchment and sub-catchment scales. This finding generally holds true when evaluating both early (e.g., 1971, 1988, 1992) and late (e.g., 1972, 1991) spring freshet seasons, and both the westerly oriented Dennis Creek sub-catchment and the southerly oriented 240 Creek and 241 Creek sub-catchments. In addition, the observed snowpack data were matched well regardless of elevation, solar exposure, or vegetation type, and regardless of the seasonal snowpack accumulation being high (e.g., 2011-2013) or low (e.g., 2009-2010 & 2014).

Notwithstanding the good overall fit, the Dennis Creek spring freshet peaked and receded somewhat too early in 1989 and 1991 (Fig. S3.4); however, these discrepancies were small relative to the overall duration of the spring freshet, and were not unexpected considering the simulation relied on the meteorological record that was extended using regional data. For Penticton Creek and Dennis Creek low flows, the performance statistics decreased considerably from the calibration to the validation; however, these decreases were considered acceptable since winter hydrometric records can be subject to large errors (e.g., ice influences). Collectively, the calibration and validation results indicate good overall representation of the distribution of precipitation, air temperature, ET, snowpack processes, and fast and slow runoff response mechanisms.

## 3.2 Water input dynamics

### 3.2.1 Rainfall

The combined effects of the projected increase in air temperature and a shift in the seasonal distribution of precipitation for the future climate conditions resulted in increased winter and spring rainfall (based on the rain-snow partitioning simulated in the model), and decreased summer rainfall (Table 3), despite only small changes in overall winter and spring precipitation for





the 2050s (Table 1). The relative change in spring rainfall intensity increased with increasing event duration. Changes in the 2080s were +18%, +24%, and +29% for 1-, 3-, and 14-day durations (based on data in Table 3).

**Table 3: Mean rainfall (mm) for the current climate at the P1 meteorological station, and the incremental change projected for the future climates. Winter and summer values are the long-term mean of the seasonal total. Spring values are the long-term mean of the seasonal maximum for the specified duration.**

| Climate condition | Winter | Spring | | | Summer |
|---|---|---|---|---|---|
| | | 1 day | 3 days | 14 days | |
| Current | 10 | 22 | 31 | 60 | 143 |
| 2050s | +20 | +2 | +4 | +9 | -51 |
| 2080s | +77 | +4 | +7 | +18 | -54 |

### 3.2.2    Net precipitation

On an annual basis, net precipitation (precipitation minus evapotranspiration) at the high elevation (1800m) decreased under the 2050s climate, and partially recovered under the 2080s climate (Table 4). Future decreases were greater for clearings than forests, particularly on north-facing sites. For the winter, net precipitation increased under the 2080s climate. For the summer, net precipitation decreased substantially under the 2050s climate, but partially recovered in the 2080s (except for north-facing clearings).

**Table 4: Mean of net precipitation (precipitation minus evapotranspiration) (mm) by slope aspect, vegetation type, season, and climate condition for the high elevation (1800m) (incremental change compared to the current climate is provided in parentheses). The south- and north-facing forested sites are moderate density lodgepole pine and spruce-fir stands, respectively. These are the most common stand types for each slope aspect at this elevation.**

| Season | Climate condition | South | | North | |
|---|---|---|---|---|---|
| | | Clearing | Forest | Clearing | Forest |
| Annual | Current | 535 | 408 | 605 | 476 |
| | 2050s | 418 (-117) | 325 (-83) | 476 (-129) | 369 (-107) |
| | 2080s | 452 (-83) | 357 (-51) | 500 (-106) | 414 (-63) |
| Winter | Current | 190 | 174 | 190 | 181 |
| | 2050s | 191 (+1) | 173 (-1) | 191 (+1) | 182 (+1) |
| | 2080s | 220 (+30) | 198 (+24) | 220 (+30) | 209 (+28) |
| Summer | Current | -54 | -102 | -15 | -78 |
| | 2050s | -164 (-110) | -168 (-67) | -145 (-130) | -178 (-100) |
| | 2080s | -150 (-96) | -143 (-42) | -153 (-138) | -151 (-73) |



### 3.2.3 Snowpack accumulation

#### 3.2.3.1 Influence of topography and vegetation under current climate

Snowpack accumulation under the current climate varied directly with elevation and inversely with solar exposure (Fig. 4a & 5a). Clearings generally accumulated more snow than forests, which is a manifestation of the disturbance effect that includes influences on both snowpack accumulation and melt. From the site scale snowpack analysis, the difference in annual maximum SWE between forests and clearings ranged from 4 to 87 mm (2% and 32% lower in forests, respectively) (Fig. 4b). The disturbance effect was greater for higher stand densities (as represented through LAI), and on sites with high solar exposure. From the catchment scale snowpack analysis, the greatest disturbance effect was 123 mm (35% lower in forest), which was in a southerly facing high density Pl stand in the ESSFL zone (~1660 m elevation) (Fig. 5d).

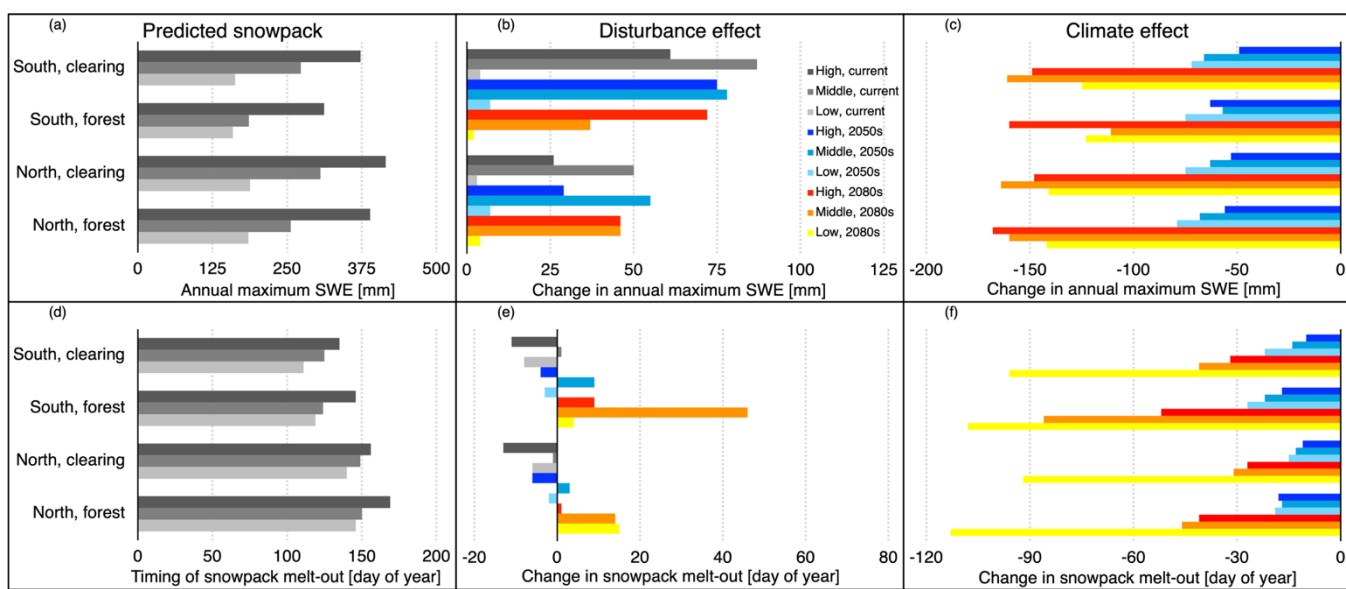

**Figure 4: Mean of annual maximum SWE (mm) and median timing of snowpack melt-out (day of year) by slope aspect, stand type, and elevation for the current climate (panels 'a' and 'd'); and the incremental change predicted for the disturbance effect (panels 'b' and 'e') and the climate effect (panels 'c' and 'f') (output from site scale snowpack sensitivity analysis). The most common stand type is represented for each slope aspect and elevation, including low density P/F at the low elevation, moderate density P/F at the middle elevation, and moderate density P/F (south aspect) and S/B (north aspect) at the high elevation.**





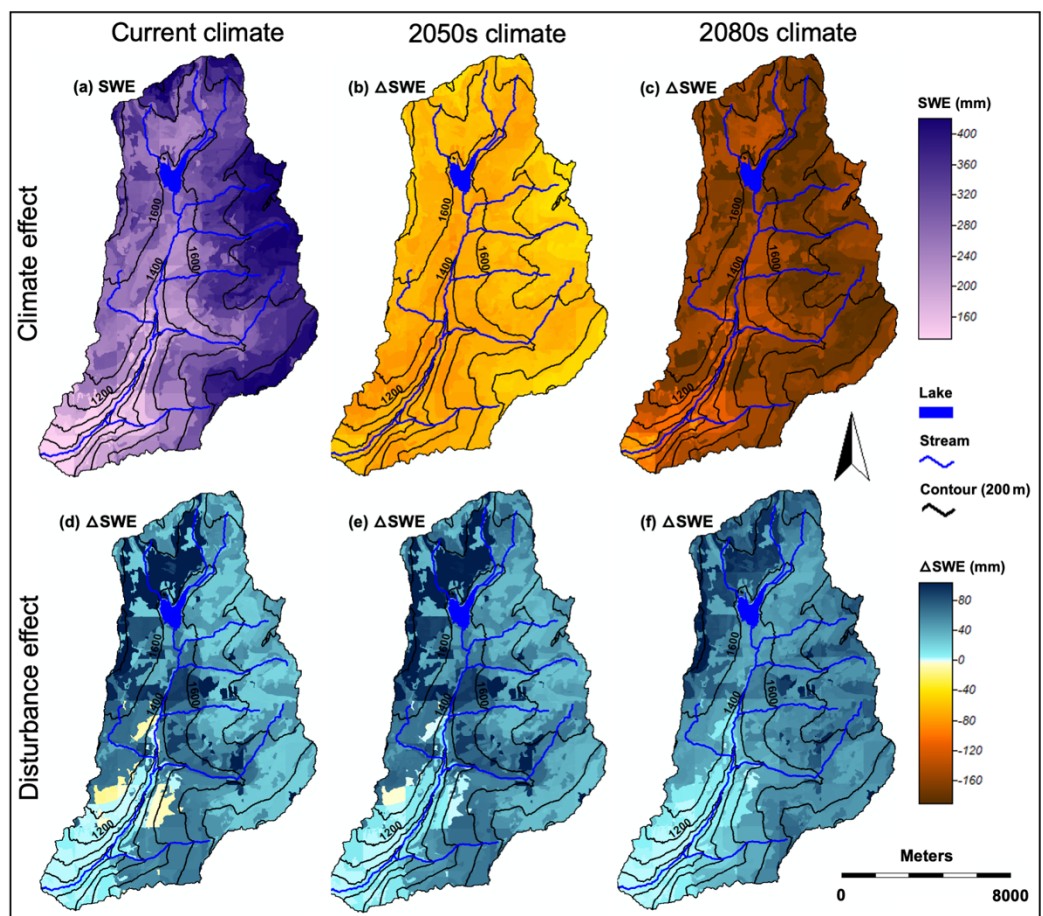

**Figure 5: Distribution of annual maximum SWE for the mature forest condition (panel 'a'), and the change in annual maximum SWE for the mature forest condition under the 2050s (panel 'b') and 2080s (panel 'c') climates (i.e., climate effect). For each climate condition, the corresponding disturbance effect (panels 'd'-'f') shows the difference in annual maximum SWE for the 100% cleared condition compared to the mature forest condition.**

### 3.2.3.2 Combined influences of climate and landcover change

Combining all site conditions simulated in the site scale snowpack analysis, the overall mean of the annual maximum SWE was predicted to decrease by 65 mm (-24%) and 146 mm (-55%) under the 2050s and 2080s climates, compared to the current climate, respectively (i.e., climate effect). Decreases were predicted for all site conditions, regardless of elevation, solar exposure, or vegetation type (Fig. 4c & 5b-c). The climate effect was generally greater at the low elevation for the 2050s, but greater at the middle and high elevations for the 2080s (Fig. 4c).

Clearings still generally accumulated more snow than forests under the future climate conditions, for all elevations. Generally, the disturbance effect increased at the high elevation under a changing climate, and decreased at the middle elevation (Fig. 4b



& 5d-f). The greatest disturbance effect in the 2050s was on south-facing sites at the middle and high elevations, and on south-facing sites at the high elevation in the 2080s (Fig. 4b).

### 3.2.4 Snowmelt timing

#### 3.2.4.1 Influence of topography and vegetation under current climate

The timing of snowpack melt-out under the current climate varied directly with elevation, and inversely with solar exposure (Fig. 4d & 6a). Sites with high solar exposure melted out earlier than sites with low solar exposure, ranging from 27-29 days earlier at the low elevation, to 14-21 days earlier at the high elevation (larger values correspond to forests). Clearings generally melted out earlier than the corresponding mature forests (i.e., disturbance effect). In the site scale analysis, clearings melted out 6-8 days earlier at the low elevation, and 11-13 days earlier at the high elevation (Fig. 4e). This disturbance effect was minimal at the middle elevation for the vegetation types represented in the site scale analysis (i.e., moderate density P/F; Fig. 4e); however, based on the catchment scale analysis, large disturbance effects were associated with high density stands located in the MS zone at the middle elevation (Fig. 6d).





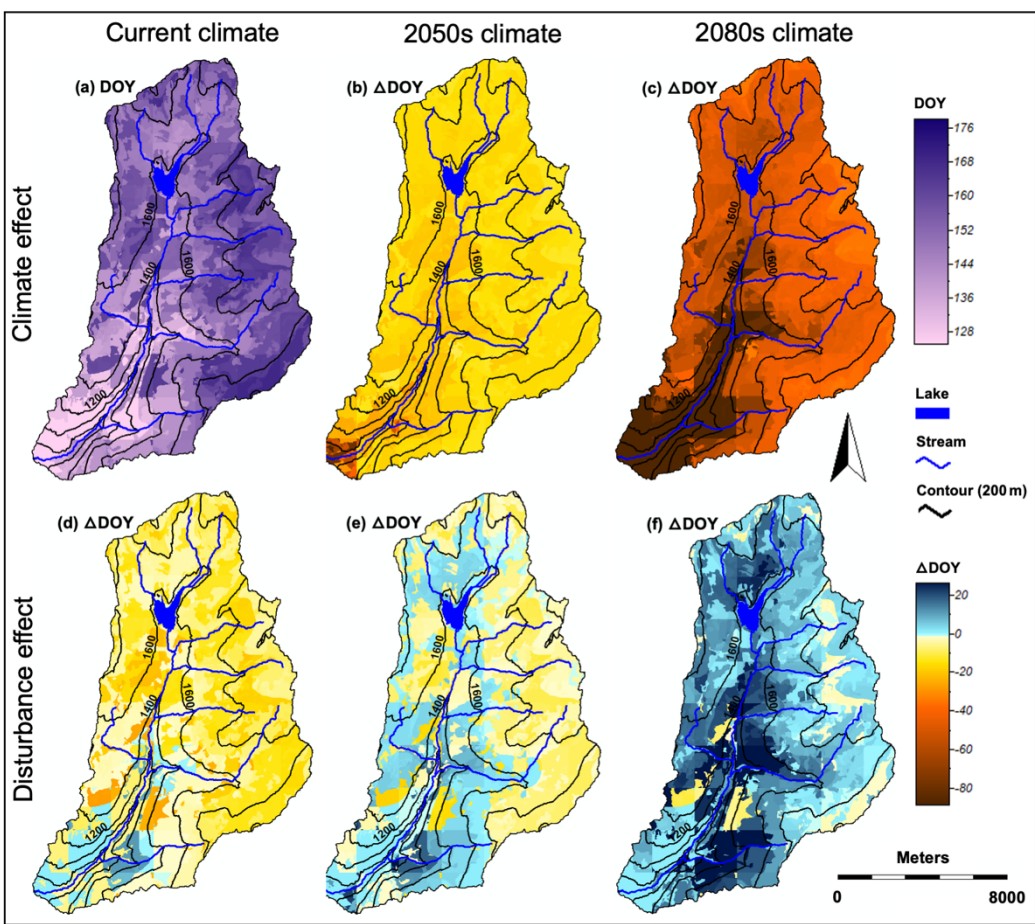

**Figure 6: Distribution of melt-out timing (day of year, DOY) for the mature forest condition (panel 'a'), and the change in melt-out timing for the mature forest condition under the 2050s (panel 'b') and 2080s (panel 'c') climates (i.e., climate effect). For each climate condition, the corresponding disturbance effect (panels 'd'-'f') shows the difference in melt-out timing for the cleared condition compared to the mature forest condition.**

### 3.2.4.2   Combined influences of climate and landcover change

With each successive climate condition, the timing of peak snowpack accumulation advanced, and the duration of the melt phase was extended due to an increase in the persistence of midwinter melt caused by higher air temperatures (Fig. S4). These changes were more persistent for the 2080s climate, with an associated upward shift in the elevations experiencing midwinter melt.

Combining all site conditions simulated in the site scale snowpack analysis, the overall mean timing of snowpack melt-out was predicted to advance by 17 days under the 2050s climate, and 64 days under the 2080s climate (i.e., climate effect). The timing was predicted to advance for all site conditions, regardless of elevation, solar exposure, or vegetation (Fig. 4f & 6b-c).



The climate effect was greatest for lower elevations and higher stand densities (compare Fig. 6b-c to Fig. 2d). In the 2080s, high solar sites generally experienced a larger climate effect than low solar sites (Fig. 4f).

The disturbance effect (i.e., comparing forests and clearings) generally shifted under climate change from advancing to delaying the timing of melt-out (Fig. 6d-f), particularly for south-facing sites at the middle and high elevations (Fig. 4e). This

response caused clearings to melt out later than forests under the 2080s climate.

### 3.3 Runoff dynamics

### 3.3.1 Timing

The long-term simulations predicted an advance in the timing of the spring freshet (both rising and falling limbs) with each successive climate condition, and an elongation of the hydrograph in the 2080s (Fig. 7b). For the forested landcover condition,

the mean date of the annual maximum discharge (i.e., peak flow) was day 151 of the year under the current climate, and advanced by 18 days and 37 days in the 2050s and 2080s, respectively. Under the 2080s climate, discharge was predicted to rise during the winter, particularly during February and March.

Under the current climate, the timing of the peak flow advanced by a mean of seven days for the large burn, and the hydrograph

elongated somewhat (i.e., disturbance effects) (Fig. 7a); however, these effects diminished under a changing climate (Fig. 7b). Changes in hydrograph timing were smaller for the other landcover conditions (Fig. 7a).





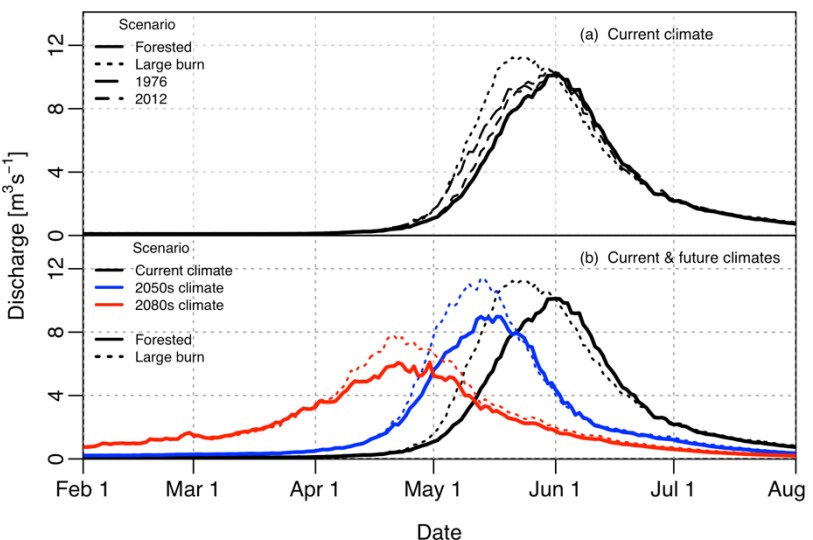

**Figure 7: Median of daily discharge from the long-term simulations for several landcover conditions under the current climate**
**(small burn condition excluded for clarity, as it coincides closely with the forested condition), and for the forested and large burn**
**landcover conditions under all climates.**

### 3.3.2 Peak flow

Compared to the forested condition, the peak flow increased for all disturbed landcover conditions (i.e., disturbance effect), particularly for extreme events, and regardless of climate (Fig. 8a-b & 9a-b; Table S4). For events with a 100 year return period (i.e., 1% probability of occurrence in a given year), the large burn disturbance effect was a peak flow increase of 18%, 15%, and 11% for the current, 2050s, and 2080s climates, respectively (i.e., same large burn landcover condition under three different climate conditions). Similarly, the disturbance effect for 100 year events was an increase of 6%, 9%, and 9% for the 1976 landcover condition, and 1%, 4%, and 3% for the 2012 landcover condition, respectively. Changes were less than 1% for the small burn.

For all landcover conditions, the peak flow decreased under a changing climate for frequently occurring peak flows (<5-10 years for the 2050s; <20-25 years for the 2080s), but increased for extreme events (i.e., climate effect) (Fig. 8b & 9c). The climate effect for 100 year events was an increase of 18% and 22% in the 2050s and 2080s for the forested condition, respectively. Expressed as a frequency shift, a 22.9 m3/s peak flow would have a return period of 100, 33, and 39 years under each successive climate condition for the forested condition, respectively. A 22.9 m3/s peak flow would have a return period of 16, 11, and 20 years under each successive climate condition for the large burn.



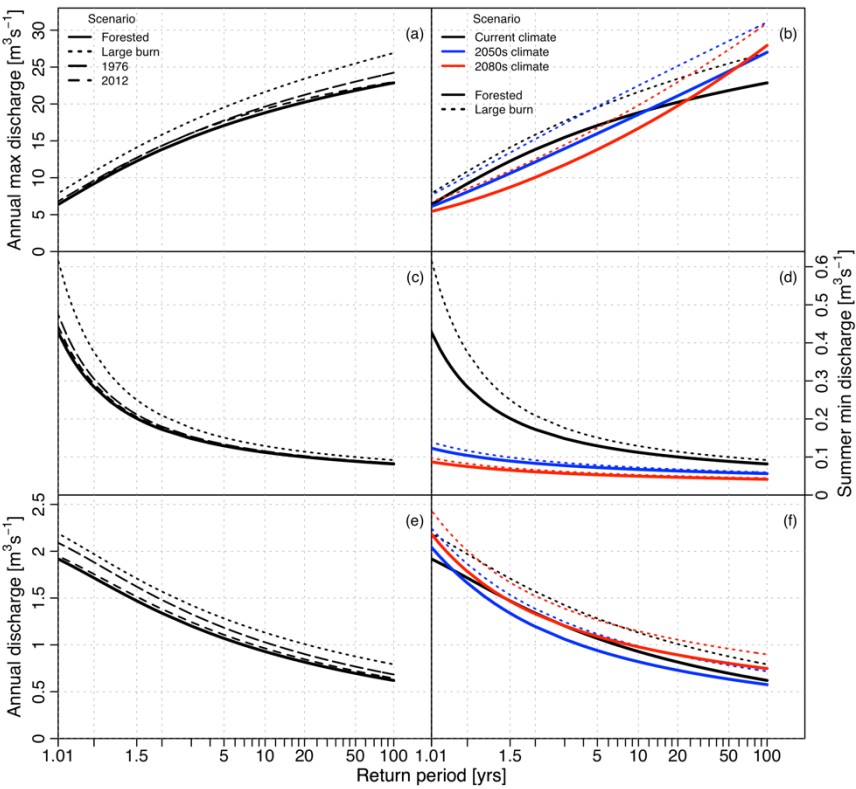

**Figure 8: Event frequency curves from the long-term simulations for the current climate (panels 'a', 'c' & 'e'), and for the forested and large burn landcover conditions under all climates (panels 'b', 'd' & 'f'). Small burn condition excluded for clarity (coincides closely with the forested condition).**







**Figure 9: Discharge for 2 and 100 year return periods by landcover condition and event frequency for the current climate (i.e., predicted discharge) (panels 'a', 'd' & 'g'), and the incremental change predicted for the disturbance effect (panels 'b', 'e' & 'h') and the climate effect (panels 'c', 'f' & 'I'). Values estimated through the event frequency analyses on output from the long-term simulations (Fig. 8). Small burn condition excluded for conciseness (coincides closely with the forested condition).**

### 3.3.3   Summer low flow

For all climate conditions, the disturbance effect increased summer low flows across all return periods for the large burn condition, with intermediary increases for the 1976 condition (Fig. 8c-d & 9d-e; Table S4). Changes were small for the 2012 and small burn landcover conditions. For 2 year events, the large burn disturbance effect was a low flow increase of 21%, 10%, and 8% for the current, 2050s, and 2080s climates, respectively. The disturbance effect for 2 year events was an increase of 3-4% for the 1976 condition, and 1-2% for the 2012 and small burn conditions, regardless of climate.

For all landcover conditions, low flows decreased substantially under a changing climate for all return periods (Fig. 8d & 9f). The climate effect for 2 year events was a decrease of 52% and 65% in the 2050s and 2080s for the forested condition,


respectively. Expressed as a frequency shift, a 0.092 m3/s low flow would have a return period of 100, 2.0, and 1.0 years under
each successive climate condition for the large burn condition, respectively. A 0.092 m3/s low flow would have a return period
of 37, 1.5, and 1.0 years under each successive climate condition for the forested condition.

### 3.3.4    Annual discharge

For all climate conditions, the disturbance effect increased annual discharge across all return periods for the large burn
condition, with intermediary increases for the 1976 condition (Fig. 8e-f & 9g-h; Table S4). Changes were small for the 2012
and small burn landcover conditions. For 2 year events, the large burn disturbance effect was an annual discharge increase of
17%, 16%, and 14% for the current, 2050s, and 2080s climates, respectively. The disturbance effect for 2 year events was an
increase of 10-11% for the 1976 condition, 3% for the 2012 condition, and less than 1% for the small burn, regardless of
climate.

For all landcover conditions, annual discharge decreased under the 2050s climate (compared to the current climate) for all
return periods, but fully recovered under the 2080s climate for almost all return periods (Fig. 8f & 9i). The climate effect for
2 year events was a decrease of 11% and 1% in the 2050s and 2080s for the forested condition, respectively. Expressed as a
frequency shift for low annual discharge, a 0.79 m3/s annual discharge would have a return period of 100, 46, and >100 years
under each successive climate condition for the large burn condition, respectively. A 0.79 m3/s annual discharge would have
a return period of 24, 12, and 57 years under each successive climate condition for the forested condition (i.e., low annual
discharge is more frequent under the forested condition).

### 3.4    Conditions during extreme events

### 3.4.1    Peak flow

Figure 10 shows hydrometeorological conditions for the largest peak flow event under each climate condition. The peak flow
increased under each successive climate condition, with values for the forested condition of 25.2, 28.1, and 30.2 m3/s under
the current, 2050s, and 2080s climates, respectively (long-term mean under the current climate was 14.0 m3/s) (Fig. 10g-i).
The difference in peak flow between the forested and large burn conditions decreased under each successive climate condition,
with corresponding values of +4.1, +3.4, and +1.8 m3/s.






During the 4-6 days prior to and during these peak flow events, precipitation generally increased with each successive climate condition, particularly for the 2080s (Fig. 10a-c). Daily low temperatures at the P1 weather station (1619 m elevation) generally remained above 0°C, indicating that most or all precipitation within the catchment fell as rain for the events. The snowpack loading generally decreased and the extent of snow-free terrain during the peak flow increased with each successive climate

condition (Fig. 10d-f).

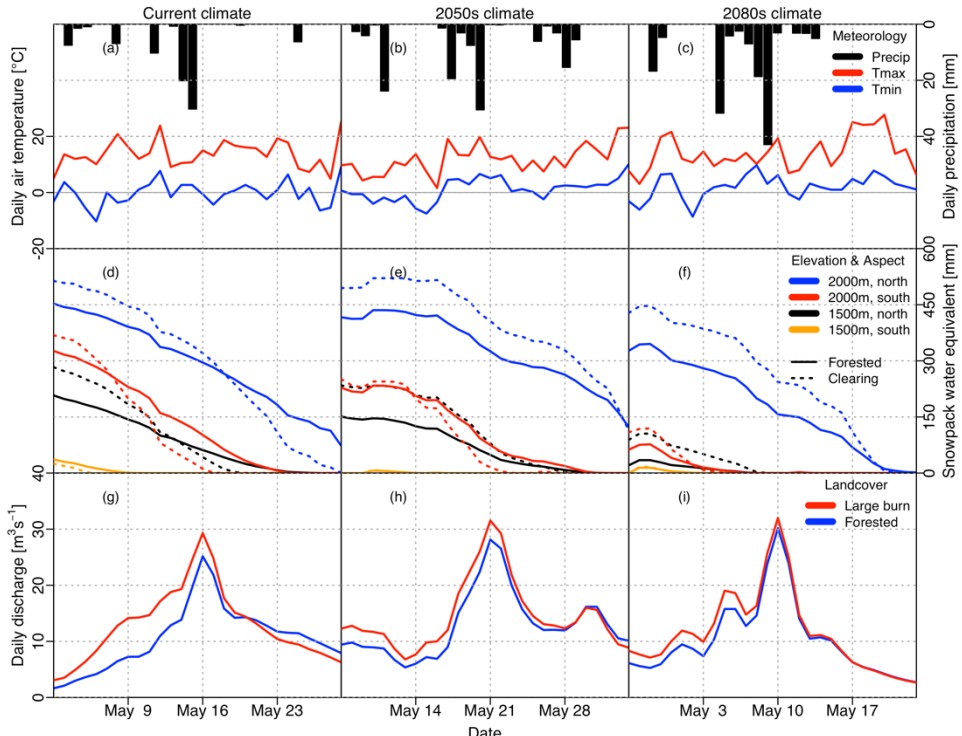

**Figure 10: Hydrometeorological conditions (daily data) for the largest peak flow event under each climate condition. For the SWE data (i.e., panels 'd' through 'f') solid lines are clearings, 2000m dashed lines are moderate density spruce-fir stands, and 1500m dashed lines are moderate density lodgepole pine stands. These are the most common stand types for each combination of elevation**
**and aspect.**

### 3.4.2    Summer low flow

Figure 11 shows hydrometeorological conditions for the lowest summer low flow event under each climate condition. The low flow decreased under each successive climate condition, with values for the forested condition of 0.076, 0.055, and 0.040 m3/s
under the current, 2050s, and 2080s climates, respectively (long-term mean under the current climate was 0.19 m3/s) (Fig. 11d).





Prior to these events, net precipitation for each month was within ~50 mm of normal under the current climate, except in May
and June when net precipitation was 50-120 mm below normal for the 2050s and 2080s climates (Fig. 11a). For all climates,

peak snowpack accumulation was at or below normal and the snowpack melted early (Fig. 11c). These conditions generated
peak flows that were slightly below normal under the current climate, and normal under the 2050s and 2080s climates;
however, the recession flow began approximately one and two months earlier than normal in the 2050s and 2080s, respectively,
resulting in lower summer flows (Fig. 11d).

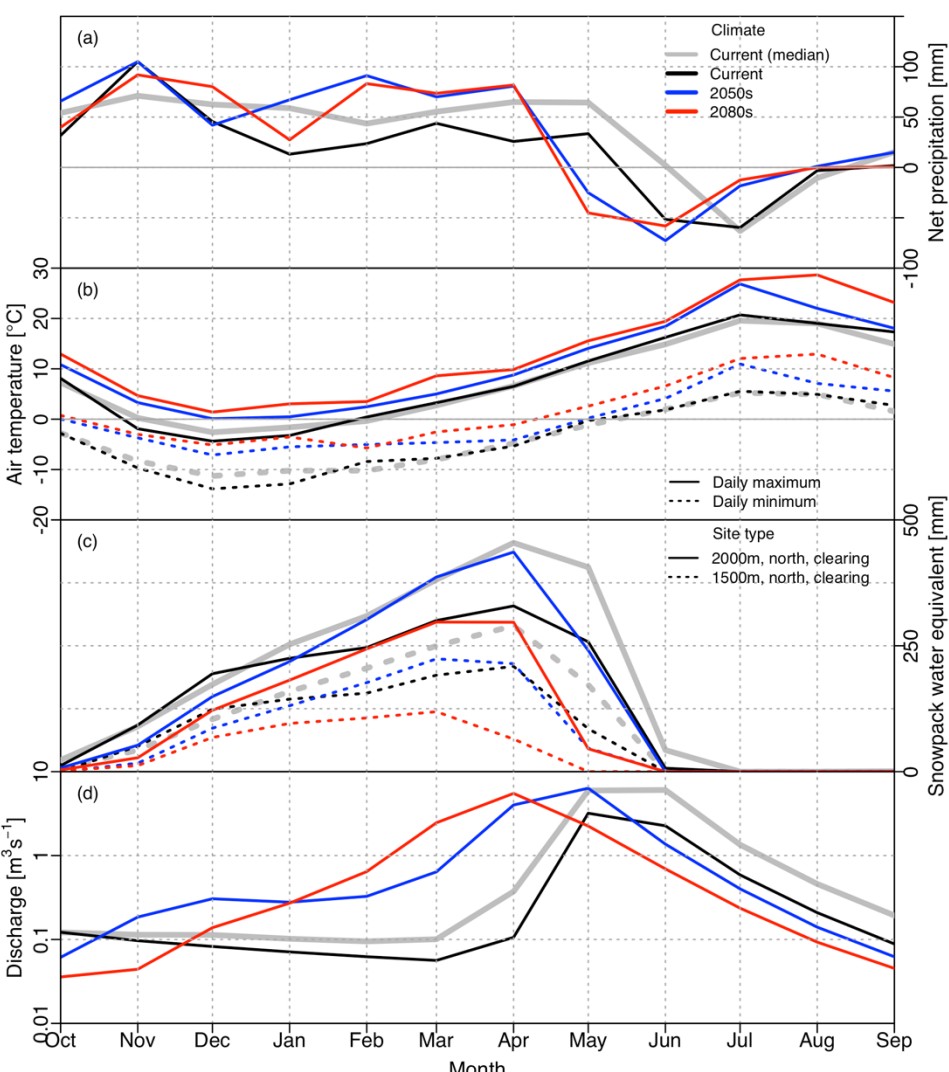

**Figure 11: Hydrometeorological conditions (monthly data) for the year with the lowest summer low flow under each climate
condition. Panels 'a' though 'c' represent meteorological conditions at the P1 station. Panel 'd' shows the hydrographs for the
forested landcover condition. The grey line provides the long-term median value for each month under the current climate as a
baseline for comparison. Note the logarithmic y-axis in panel 'd'.**



### 3.4.3    Annual discharge

Figure 12 shows hydrometeorological conditions for the lowest annual discharge event under each climate condition. The annual discharge values for the forested condition were 0.57, 0.42, and 0.70 m3/s under the current, 2050s, and 2080s climates, respectively (long-term mean under the current climate was 1.32 m3/s) (Fig. 12d).

During these events, net precipitation was at or below normal in almost all months (Fig. 12a), and peak snowpack accumulation was well below normal (Fig. 12c). These conditions generated spring freshet flows that were well below normal under all three climate conditions; however, discharge was substantially above normal between November and April for the 2080s climate, particularly in March and April (Fig. 12d). These elevated flows for the 2080s were associated with above normal air temperatures throughout the winter (Fig. 12b), and near normal net precipitation in February, March, and April, which would have resulted in higher rates of mid-winter runoff from snowmelt and rainfall.





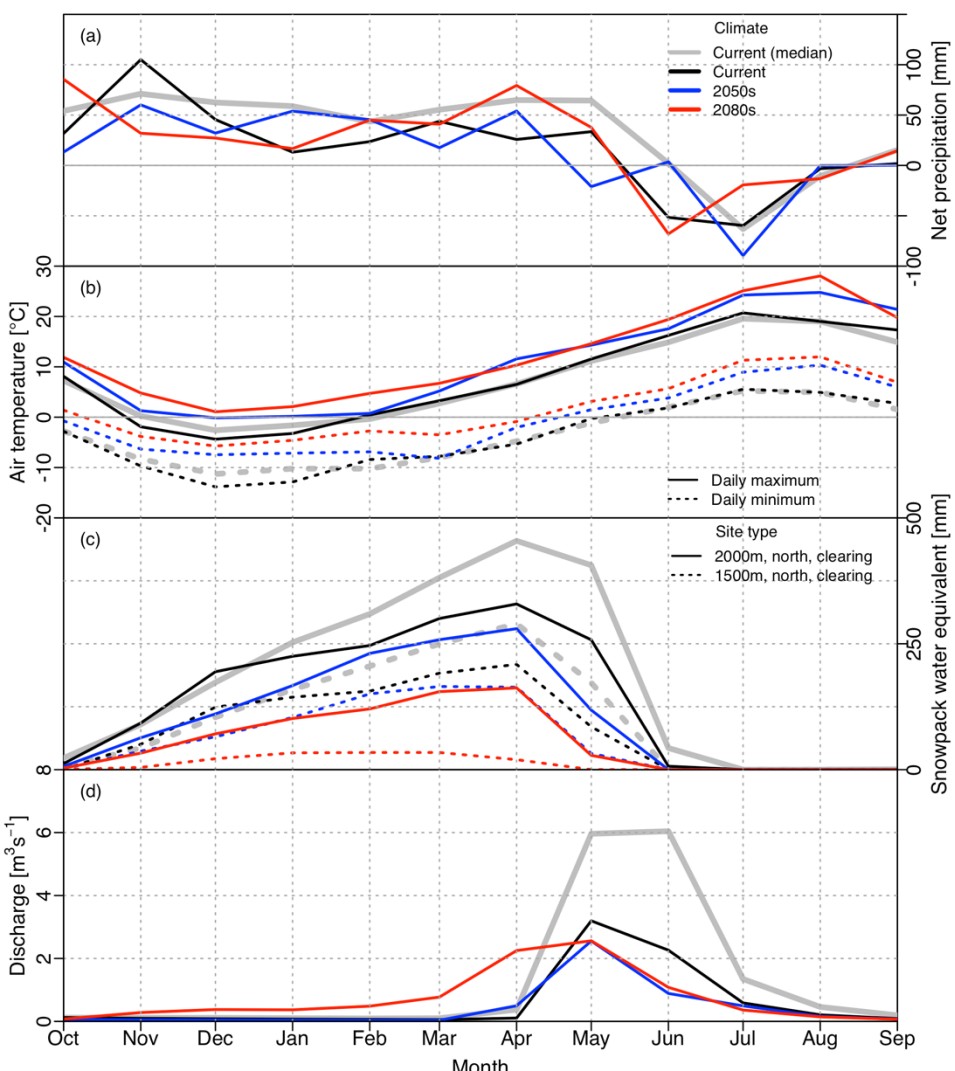

**Figure 12: Hydrometeorological conditions (monthly data) for the year with the lowest annual discharge under each climate condition. Panels 'a' though 'c' represent meteorology at the P1 station. Panel 'd' shows the hydrographs for the forested landcover condition. The grey line provides the long-term median value for each month under the current climate as a baseline for comparison.**

## 4    Discussion

### 4.1    Implications to flooding

The finding that stand replacing disturbance can increase peak flows for a period of time after disturbance is well established in the scientific literature (Goeking and Tarboton, 2020). The modelling in the current study predicts that the combined impacts on peak flows of stand replacing disturbance and climate change are generally offsetting for events with return periods less





than 5-25 years (i.e., disturbance increases yield, climate change decreases yield), but additive for extreme events (i.e., disturbance and climate change both increase yield) (Sect. 3.3.2). However, stands in advanced stages of regeneration (i.e., 2012 condition) were found to be highly effective at mitigating the influences of disturbance on peak flows.

For the nearby Upper Similkameen River Watershed, Li et al. (2018) found landcover disturbance and climate variability to
have offsetting influences on the volume of annual surface runoff (i.e., quick flow only; excludes baseflow) and annual streamflow (i.e., total flow), averaged over a 14 year period (for comparison, a long-term mean flow typically has a return period of approximately 2 years). Offsetting influences were also found for other watersheds in the BC Interior (Wei and Zhang, 2010; Zhang and Wei, 2012). However, Giles-Hansen et al. (2019) found landcover disturbance and climate variability to have additive influences on discharge for the nearby Deadman River Watershed. They suggested that the additive influence
might be associated with the disturbance period being relatively wet (i.e., above normal flows). The findings from these studies corroborate the current study with respect to a shift from offsetting to additive influences for increasing event magnitude.

The results discussed above for the current study are generated by a complex interplay between landcover change and climate change with respect to impacts on canopy interception processes, snowpack dynamics, and the resulting runoff regime. This
interplay is interwoven with non-linear runoff response behaviour influenced by changing precipitation intensities, rain-snow partitioning, and the distribution of forest cover density. This complexity is discussed below.

### 4.1.1 Influence of landcover distribution

The flood frequency results highlight a dependency of extreme peak flows on the distribution of landcover disturbance rather than strictly the amount of disturbance. This finding is supported by the much greater disturbance effect on peak flows under
the current climate for the large burn compared to the 1976 condition (both had disturbance covering ~1/3 of the catchment) (Sect. 3.3.2). Catchment SWE increases under both landcover conditions, with greater increases for the large burn due to the disturbance being at higher elevations where snowpack accumulation and stand densities are greater (Fig. 2d). A larger increase in catchment SWE for the large burn (Sect. 3.2.3.1) helps explain the greater disturbance effect on peak flows; however, changes in the synchronization of runoff timing between higher and lower elevations likely also played a role. In this respect,
a post-disturbance advance in the timing of snowpack melt-out in the lower and middle elevations (i.e., early melting areas) (1976) (Sect. 3.2.4.1) generates antecedent conditions with drier soils and less snowpack during the peak flow period, and, thus, results in a lower contribution to the peak flow (i.e., desynchronization). In contrast, an advance in the timing of snowmelt at higher elevations (i.e., late melting areas) (large burn) synchronizes the timing of runoff with downslope areas during the peak flow. Increased synchronization could increase the peak flow over and above that caused by strictly increasing catchment
SWE, and vice versa for desynchronization (Bewley et al., 2010; Ellis et al., 2013; Pomeroy et al., 2012; Winkler et al., 2015).





A comparison between the large burn, 1976, and 2012 conditions further highlights the influence of landcover distribution on runoff, whereby a catchment disturbance level of 47% (2012) generated little or no disturbance effect on peak flows under the current climate (Sect. 3.3.2). The 2012 condition incorporated forest stands in various stages of regeneration (Fig. 3e); thus, the 2012 condition illustrates hydrologic recovery of the disturbance effect through recovery in the rates of evapotranspiration and, likely, the snowmelt energy budget. For UPC and Mayson Lake, Winkler and Boon (2015) showed ~80% recovery in the disturbance effect on snowpack ablation and accumulation for regenerating lodgepole pine stands at approximately 50-60% of the mature height. Also for UPC, Winkler et al. (2021) showed more-or-less full recovery of the soil moisture content in the root zone of regenerating lodgepole pine stands at approximately 10% of the mature height.

### 4.1.2    Influence of climate change

The results illustrate an increasing importance of rainfall in controlling peak flow response under a changing climate, at the expense of snowmelt influence (Sect. 3.4.1) (Merritt et al., 2006; Whitfield et al., 2002). This finding is supported by three points: (1) the increasing persistence of midwinter rainfall and snowmelt; (2) the decreasing magnitude of frequently occurring peak flows that correspond to decreasing snowpack loads; and (3) the increasing magnitude of extreme peak flows associated with increasing spring rainfall intensity. The latter outcome occurred even with an increase in the extent of snow-free terrain during the peak flow, further highlighting the increasing influence of rainfall in controlling runoff (e.g., rain on snow, rain on wet soils).

### 4.1.3    Combined influences of landcover and climate

The combination of climate change and the large burn condition is predicted to advance the timing of the peak flow three to five times more than the advance generated by the large burn condition alone (Sect. 3.3.1), showing a dominant effect of climate on the timing of spring freshet. With respect to the magnitude of extreme peak flows, the modelling predicts a similar influence for landcover disturbance and climate change. However, the results also suggest a decrease in the sensitivity of extreme peak flows to disturbance at high elevations under a changing climate (Sect. 3.3.2). This decreasing sensitivity is demonstrated by the decrease in the large burn disturbance effect with successive climates. Similarly, Cristea et al. (2013) found the climate effect on the timing of spring freshet runoff and peak flow yield to be greater for a forested catchment condition (dense lodgepole pine in lower 39% of basin, with alpine above) than a disturbed condition. The current study also suggests that, as rainfall runoff becomes more important for extreme peak flows under a changing climate, the specific landcover condition in the middle and low elevations will also become more important. This finding is supported by the increasing disturbance effect for the 1976 condition with successive climates (Sect. 3.3.2).

We inferred that the decreasing sensitivity to the large burn condition is related somewhat to a future decrease in the ratio of canopy interception capacity to rainfall intensity for extreme events (Sect. 3.2.1). Lewis et al. (2001) found that relative increases in discharge peaks declined with increasing rainstorm size. Furthermore, the increasing sensitivity to the 1976





condition under a changing climate is likely related to increasing catchment-wide synchronization of runoff, with rainfall
inputs occurring more-or-less simultaneously throughout the catchment (in comparison to the asynchronous distribution of
snowmelt) and extreme rainfall intensities increasing. In this respect, Asano et al. (2020) found that the peak propagation speed
of hillslope runoff increased by two orders of magnitude with increasing rainstorm size. The snowpack results also help explain
the changing sensitivities. That is, the modelling predicts that clearings will moderate the impacts of climate change on
snowpack accumulation at the high elevation, whereas this influence is opposite or negligible at the middle and low elevations
(Sect. 3.2.3.2).

## 4.2 Implications to water supply

### 4.2.1 Summer low flow

Similar to the conclusions of Dierauer et al. (2021), our results suggest that extreme summer low flows will become
commonplace in the future, with most of the change in frequency occurring by the 2050s (Sect. 3.3.3). With respect to
influencing the severity of extreme summer low flows, the hydrometeorology conditions shown in Fig. 11d suggest that the
timing of the post-freshet recession flow is more important than the volume of the spring freshet runoff. An advance in the
recession flow would cause an earlier start to the low flow period and provide additional time for soil desiccation during the
summer, which would decrease the low flow (Dierauer et al., 2021). Notwithstanding this influence, summer low flows were
increased by the large burn under the current climate, even while generating a small advance in the timing of the recession
flow (Fig. 7a). This contrast suggests that disturbance related lower ET (increases low flow) and earlier melt-out (decreases
low flow) have unequal impacts, with the former having greater influence on the severity of low flows.

The large burn effect on the timing of the recession flow transitions from a small advance under the current climate, to a small
delay under the 2080s climate (Fig. 7b). This reversal in the disturbance effect is associated most directly with disturbance to
high density stands on a southerly exposure at the middle and high elevations (Sect. 3.2.3.2 & 3.2.4.2). We infer that this
reversal is related primarily to meteorological related changes within forests – likely decreasing snowpack accumulation and
increasing below canopy net long-wave radiation associated with increasing air temperature (Cristea et al., 2013). These
conditions would generate more frequent midwinter melt in forests. Notwithstanding these findings, the decreased sensitivity
of the low flow volume to landcover condition (with up to 1/3rd of a catchment disturbed) under a changing climate suggests
that the influence of stand replacing disturbance will likely decrease. To exemplify this point, the large burn increased the 2
year low flow by 0.037 m3/s under the current climate, whereas the increase was only 0.0051 m3/s under the 2080s climate
(Fig. 9e).



### 4.2.2 Annual discharge

Low annual discharge is predicted to become more prevalent by the 2050s, but then fully recover or become less prevalent

(compared to the current climate) by the 2080s because of increased precipitation in the fall-spring period (Table 1) (Rasouli et al., 2014). Rasouli et al. (2019) found that reduced snowpack sublimation and increased precipitation offset the impact of increased summer ET, causing annual yield to be maintained under a changing climate. The current study also suggests that landcover disturbance can have a mitigative influence on low water supply that is predicted to be sustained under a changing climate for annual discharge, but minimally for low flow (Fig. 9e & 9h). A low annual discharge occurring, on average, once

in a 100 year period under the large burn condition, would occur four to five times under the forested condition, regardless of climate (Fig. 8f). This mitigative influence is generated primarily by increased spring freshet runoff related to reduced winter ET – an influence that diminishes as forest stands reach advanced stages of regeneration (e.g., 2012 condition) (Fig. 8e).

For the Penticton Creek Watershed, the snowpack analyses suggest that the largest disturbance related increases in annual

runoff yield likely occur with high density stands at the middle and high elevations, particularly high density stands in the high elevation with a southerly exposure (Sect. 3.2.3). This response is predicted to increase under a changing climate for the high elevation, whereas the snowpack and runoff analyses suggest that the response to disturbance in the low and middle elevations will not vary substantially with climate.

Winkler et al. (2017) found that increases in annual yield within the 241 Creek sub-catchment averaged only 5% after 47% of the catchment was logged, compared to an increase of 17% for the large burn in the current study for a 2 year event under the current climate. Possible reasons for this discrepancy include large differences in catchment scale, and differences in the distribution of disturbance relative to stand density. In particular, the highest density stands in the Penticton Creek Watershed, which showed the greatest disturbance effects, do not generally coincide with the 241 Creek sub-catchment (compare Fig. 1,

2d, and 5d). The original stands in the 241 Creek sub-catchment had pre-dominantly moderate densities (Fig. 2d).

### 4.2.3 Interplay between low flow & annual discharge

The advance in the recession flow (Fig. 7b) is a partial explanation for why climate change is predicted to decrease low flows severely, but not annual discharge (Cristea et al., 2013). The earlier recession flow also explains why landcover disturbance can increase annual discharge without increasing low flow. That is, increased spring freshet yield from landcover disturbance

(i.e., reduced winter ET) does little to increase low flow if the timing of the recession flow is advanced substantially by climate change (Sect. 3.4.2 & 4.2.1). For comparison, Mirmasoudi et al. (2019) found climate change increased total spring water supply by 35-39%, and decreased summer water supply by 36-79%. Dierauer et al. (2018) found that warm winters (consistent with an earlier recession flow) correspond to longer, more severe summer low flows.





Another reason for the greater influence of climate change on low flow than annual discharge is likely the decrease in summer net precipitation (Table 4), driven by increasing air temperature combined with changes in the seasonal distribution of precipitation (Table 1). Hale et al. (2022) found that a warming related shift from spring snowmelt runoff toward winter rainfall runoff resulted in an increase in annual yield, even without a change in the seasonal distribution or amount of precipitation, but also increased rates of summertime drying.

## 4.3      Managing watershed risk

When approaching watershed management from a hydrologic risk evaluation / risk management perspective, the study results indicate there is a need to carefully evaluate the interplay among environmental variables, the landscape, and the values at risk. There is potential for the same management strategy to cause offsetting and additive influences on different risks. For instance, under the current climate, the large burn (i.e., stand replacing disturbance) increased peak flows (increasing risk), increased

annual discharge (decreasing risk), and increased low flows (decreasing risk), generating offsetting influences on risk. Climate change decreased low flows and the large burn increased low flows, generating offsetting influences; however, climate change increased peak flows for extreme events and the large burn increased peak flows regardless of climate, generating additive influences.

The study results also indicate that management of hydrological risks should take a long-term perspective (e.g., 40-100 years) in temperate forest ecosystems because of the slow growth of trees, the changing climate, the variable effects of landcover disturbance on hydrology, and their interactions. For instance, in contrast to the large burn, the 2012 landscape showed little hydrological effect from 47% of the catchment experiencing disturbance spanning a 40+ year period (Fig. 9b,e,h). Moreover, the sensitivity of extreme peak flows to stand replacing disturbance was predicted to decrease under a changing climate for

disturbance at the high elevation, but increase for disturbance at the low elevation due to the influence of rain (Sect. 4.1.2 & 4.1.3). These latter findings are counter to the current risk management strategy of protecting high elevation forests and harvesting in lower elevation areas.

Having no plan or only a short-term plan for managing landcover (e.g., a prohibition on forest management, or a 5 year plan)

is not helpful with managing long-term hydrologic risk, for multiple reasons. There is potential for uncontrolled landcover disturbance (e.g., extreme wildfire, forest pests) that might (1) increase extreme flooding from widespread stand replacing disturbance (near-term impact) (Goeking and Tarboton, 2020), (2) decrease annual discharge and summer low flows from non-stand replacing disturbance (near-term impact) (Adams et al., 2012; Biederman et al., 2015; Goeking and Tarboton, 2020), and/or (3) decrease annual discharge and summer low flows when a state of dense immature forest cover is reached many

years after widespread stand replacing disturbance (future impact) (Crampe et al., 2021; Perry and Jones, 2017; Segura et al., 2020). Short-term management plans do not allow the full spectrum of hydrologic risks to be managed holistically.





The study results also show that the spatial distribution of mitigation strategies needs to be considered because synchronization and desynchronization of snowmelt timing can either exacerbate or mitigate peak flow risk. This finding is supported by two points: (1) different peak flow responses to different landcover distributions having similar overall disturbance levels (Sect. 4.1.1); and (2) variation in the disturbance effect on the timing of snowpack melt-out in relation to elevation, solar exposure, and pre-disturbance stand density (Sect. 3.2.4). This finding is supported by other synchronization related studies (Bewley et al., 2010; Ellis et al., 2013; Pomeroy et al., 2012; Winkler et al., 2015; Zhao et al., 2021).

Key pieces of the risk management toolset include wildfire management (suppression, fuel reduction, and prescribed burning) and forest management (harvesting, forest health treatments, adaptive planting, diversification). However, risk evaluation and management planning should also consider other mitigation options, including the capacity of upland reservoir storage, the volume and timing of water consumption, and municipal infrastructure. In particular, reservoir storage is likely to become more important for maintaining water supply under a changing climate because of increasing extreme peak flows, earlier spring freshet, and more severe summer drought.

We believe these results are best generalized to snowmelt-dominated montane catchments having variable slope aspects and substantial elevation relief (e.g., >500 m).

### 4.4 Uncertainty

The modelling did not incorporate uncertainty analyses addressing model parameterization or meteorological inputs, other than generating different results for the 2050s and 2080s climate conditions. With respect to parameterization, the study findings rely on the quality of the model fit as it relates to model performance (Sect. 3.1). In addition, model parameter values were assumed to remain static with climate change; thus, any long-term climate related changes in catchment processes that would substantially change runoff behaviour could result in different outcomes than those predicted in the study. For this reason, a greater level of certainty should be placed on the 2050s results than the 2080s results.

With respect to meteorology, it is inferred that climate projections with more moderate changes in air temperature (e.g., RCP4.5) would generate smaller changes in the timing and volume of spring freshet yield, and smaller decreases in summer low flows, compared to CSIRO85. Projections with more moderate changes in precipitation would likely generate smaller changes in extreme peak flows and summer low flows. In these cases, annual discharge would likely be more similar to that under the current climate. There are some RCP8.5 projections that have warmer winter and spring air temperatures, and a greater increase in precipitation, compared to CSIRO85. Under these conditions, snowmelt and the spring freshet would likely occur earlier. Greater winter and summer precipitation might offset impacts to summer low flows, and annual discharge could show a small increase.



A landcover condition with more than 1/3rd of the catchment in a recently cleared condition was not considered in this study and, therefore, the potential impacts of more widespread forest cover disturbance are unknown. Moreover, climate and landcover effects on discharge were not analyzed at different catchment scales. It is expected that impacts on event frequency would likely be more severe at smaller scales, and less severe at larger scales, because of space-time integration/averaging of
runoff processes.

The potential for wildfire to generate water repellent soil was not considered. The permeability of the soil can be severely diminished by fire, to a state where intense rainfall cannot effectively infiltrate and/or percolate, causing rapid overland flow. Under these conditions, the landcover effect on peak flows could be orders of magnitude greater than predicted in this study,
particularly over smaller scales (e.g., sub-catchment or smaller), and within the first few years after fire (DeBano, 2000).

## 5       Conclusions

The combination of climate change and stand replacing landcover disturbance in the middle and high elevations is predicted to advance the timing of the peak flow three to five times more than the advance generated by disturbance alone, showing a dominant effect of climate on the timing of spring freshet. The combined impacts of climate change and landcover disturbance
on peak flow magnitude are generally offsetting for events with return periods less than 5-25 years, but additive for more extreme events. There is a dependency of extreme peak flows on the distribution of landcover, where higher elevation disturbance has a greater effect than lower elevation disturbance. This difference is caused by greater increases in snowpack accumulation and higher stand densities at higher elevations, and changes in the synchronization of runoff timing between higher and lower elevations. However, the results also suggest a decrease in the sensitivity of extreme peak flows to high
elevation landcover disturbance under a changing climate. Moreover, stands in advanced stages of regeneration are highly effective at mitigating the influences of disturbance on peak flows.

The modelling predicts an increasing importance of rainfall in controlling peak flow response under a changing climate, at the expense of snowmelt influence. Changes include more frequent midwinter rainfall and snowmelt, decreases in the magnitude of frequently occurring peak flows corresponding to decreasing snowpack loads, and increases in the magnitude of extreme
peak flows associated with increasing spring rainfall. As rainfall runoff becomes more important for extreme peak flows under a changing climate, the specific landcover condition in the middle and low elevations is also predicted to become more important, likely because of increasing catchment-wide synchronization of runoff.






The results suggest that extreme summer low flows will become commonplace in the future, with most of the change in frequency occurring by the 2050s. The timing of the recession flow appears to be more important than the volume of runoff during spring freshet, with respect to influencing the severity of low flows. Stand replacing landcover disturbance is predicted to increase low flows, but this influence is predicted to diminish under a changing climate.


The occurrence of low annual water yield is also predicted to become more prevalent by the 2050s, but then fully recover or become less prevalent (compared to the current climate) by the 2080s because of increased precipitation in the fall-spring period. The modelling suggests that stand replacing landcover disturbance increases annual yield under current and future climates. This mitigative influence is generated primarily by increased spring freshet runoff related to reduced winter ET – an

influence that diminishes as forest stands reach advanced stages of regeneration.

The study results demonstrate the importance of a holistic approach to modelling changes to the hydrological regime, addressing snowpack and rainfall dynamics, high and low flows, extreme and frequent events, and climate and landcover changes. Moreover, for managing watershed risk, the results indicate there is a need to carefully evaluate the interplay among

environmental variables, the landscape, and the values at risk. Strategies to reduce one risk may increase others, or effective strategies may become less effective in the future. Risk management should consider many options, including wildfire management, forest management, upland reservoir storage, water consumption, and municipal infrastructure. Moreover, management of hydrological risks should take a long-term perspective (e.g., 40-100 years).

## 6      Authour contribution

Russell Smith prepared the manuscript with contributions from all co-authours. He developed the Raven catchment model, and designed and implemented the modelling investigations and analyses. Caren Dymond conceptualized the study within an overarching water risk project that was set up for the Penticton Creek Catchment, and secured funding. She also led the LANDIS-II landcover modelling. Dave Spittlehouse led the meteorological monitoring program in the Penticton Creek Watershed, and developed the synthetic weather records for current and projected future conditions. Rita Winkler led the

snowpack and hydrometric monitoring program in the Penticton Creek Watershed. Georg Jost provided an initial Raven model setup and advised on further development of the model. Caren, Dave, and Rita provided multiple comprehensive reviews of the manuscript.





## 7    Competing interests

The authours declare that they have no conflict of interest.

## 8    Acknowledgements

We would like to thank James Craig and Robert Chlumsky for their assistance with setting up and implementing Raven, as well as organizations that have supported the development of Raven (University of Waterloo, BC Hydro, Canadian Hydraulics Centre). Luke Crevier, Rachel Plewes, and Fergus Stewart provided processed spatial datasets. Barbara Zimonick and Gary

Van Emmerik aided in data collection. Craig Nitschke provided TACA model output. Sheena Spencer provided a friendly review. The City of Penticton provided information on water management in the study catchment. Funding was provided by the B.C. Ministry of Forests.

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
