# Peer review of "Modelling the effects of climate and landcover change on the hydrologic regime of a snowmelt-dominated montane catchment"

_Hydrology and Earth System Sciences, 2023_

## Author Comment (AC1)

**Author response to reviewer #1 comments for HESS manuscript "Modelling the effects of climate and landcover change on the hydrologic regime of a snowmelt-dominated montane catchment " [Paper #: hess-2023-248]**

Dear Reviewer,

We would like to thank you for your thoughtful review of the original manuscript submission. You raised several important issues that will certainly result in a stronger manuscript. Please find below a list of responses to your comments. We hope our responses satisfy the spirit and intent of your remarks.

Sincerely,

Russell Smith

**Reviewer #1 comments**

**General Comments**

I think that this contribution tackles a very important question, i.e. what is the join influence of climate and landcover change on hydrological signatures. However, similarly to the existing literature on the topic, it does not go beyond a case study. While the case study is carefully done and changes in different streamflow signature explained in detail for the one watershed under consideration, the generalizability of results is limited given that existing studies showcase the large variability of hydrologic responses to both climate and landcover changes and their interplay. In addition to not being generalizable to other regions, the results are also quite predictable given the existing literature: they point to earlier snowmelt, earlier flood peaks and an increasing influence of precipitation as we move into the future. While I do not see how the current study advances our knowledge related to future changes in streamflow signatures and the interplay between climate and landcover influences beyond the study region, I acknowledge the detailed and well-presented results for the case study watershed.

- We acknowledge that the manuscript presents a case study for a specific physiography, and that certain elements of the findings are predictable (examples you identified). We believe the results can be generalized to other snowmelt-dominated montane catchments having variable slope aspects and substantial elevation relief, and are particularly valuable to catchments with forest cover disturbance (harvesting or wildfire) and a managed water supply. There are many manuscripts addressing influences of climate change, and many others evaluating landcover;

however, the amount of literature investigating both mechanisms in detail is quite limited. Moreover, our manuscript addresses complexity in three dimensions: climate scenarios, land cover conditions, and several hydrological indicators (i.e., not strictly one or two indicators). The indicators include influences on snowpack accumulation and melt; runoff timing, magnitude, and frequency for peak flows, low flows, and annual discharge; and typical (i.e., average) and extreme events. The study examines different distributions of landcover disturbance, as well as forest regrowth. Approaching the topic in this manner revealed some predictable findings, but also findings that could only be revealed through such a holistic investigation. We will revise the abstract and introduction to highlight these points.

**Other major comments**

1. I find the methods descriptions detailed but rather superficial. That is, while the most important steps of the modeling framework are named, many methodological specificities remain unclear. A few examples:

What is the temporal resolution of the streamflow data used for the analysis? (p.4, l. 98)

- Daily data were used. This point will be clarified in the text.

Do the percentage changes in forest cover refer to the entire catchment area or just the forested catchment area (the latter would be more logical in my opinion)? (p.7, l.124-125)

- They refer to the entire catchment area. This point will be clarified in the text.
- We believe that expressing the percentage change with respect to the entire catchment more precisely relates to the proportional impact on the catchment water balance.

How were precipitation and temperature interpolated from station data to areal data? (p.7, l. 141)

- They were interpolated from the P1 weather station using lapse rates constrained by P1 and weather data from Penticton Airport near the watershed outlet. These points will be clarified in the text.

Which algorithm was used to estimate the full snowpack energy balance? (p.7, l.144)

- The snowpack balance incorporated coupled mass and energy balance equations. The full snowpack energy balance was represented using algorithms that estimate energy fluxes using daily precipitation, and daily minimum and maximum air temperature (Quick 1995). It accounted for cloud cover, short-wave radiation, long-wave radiation, and turbulent flux (Quick, 1995; Dingman, 2002). These details will be provided in the supplementary.
- Quick, M., 1995. Computer models of watershed hydrology. Water Resources Publications, Highlands Ranch, Colorado. chapter The UBC Watershed Model. pp. 233–280.
- Dingman, S., 2002. Physical Hydrology. Waveland Press Inc.

How was the historical streamflow record adjusted for storage changes in Greyback Lake? (p.8, l.152)

- Greyback Lake is a controlled reservoir (p.4, l.100). Storage in Greyback Lake was "naturalized" so-to-speak. Bathymetric data and lake level data were combined to generate a time series of daily storage change. Increasing storage was added to the discharge for the catchment outlet (and vice versa for decreasing storage), assuming an instantaneous transfer to the catchment outlet. We deem this assumption reasonable with running the model at a daily time-step, as the actual transit time during high flow periods would be ~1 hour, and the rate of storage change during low flow periods would be very low. These details will be clarified in the text, and incorporated in the discussion of uncertainty.

2. The climate impact assessment relies on one climate scenario (i.e. GCM and emission scenario combination) only, neglecting uncertainties related to emission scenario and GCM choice. While this limitation is acknowledged in the discussion section, I find that it could be overcome relatively easily by running the model for a few more climate scenarios. Furthermore, the model used for the analysis should be better contextualized within the sample of existing models (see Section 4.4.) by comparing its temperature and precipitation changes to those of other existing models.

- We acknowledge that simulating multiple climate scenarios is frequently used for projecting climate change impacts. Because of budget limitations, we had to choose between complexity of climate scenarios, land cover conditions, and hydrologic indicators. We decided to limit the climate scenarios by choosing one that had a severe climate change. Our rationale is that the scenario we chose would indicate how much

hydrology may change and, thus, pose the greatest challenge to management. In doing so, we retained complexity in land cover because it's something forest and land managers can influence. We retained complexity in the hydrological indicators because of their importance to human values. However, we plan to address your concern by running four additional climate scenarios. We will provide additional tabular and/or graphical outputs in the manuscript as a sensitivity analysis, and incorporate these results in the discussion.

3. The authors use a weather generator on the climate simulations to increase sample size (Section 2.2.2.3), which is per-se a good thing. However, it is unclear why these simulations are limited to 100-years given that the focus is among other variables on extreme events, which requires larger sample sizes to separate signal from noise.

- One hundred years of data are sufficient to estimate a 1 in 100 year event (widely considered to be an extreme event) and infer most probability distributions. Moreover, we question the validity in us generating much more than 100 years of synthetic data from 32 years of observed weather data (p.8, l.182). Any historical or climate change projection on the impact of events is limited by the historical and projected climate records. Addressing extremes that are outside of these records is an area of cutting edge research (e.g., Fischer et al. 2023, Storylines for unprecedented heatwaves based on ensemble boosting, Nature Communications, https://doi.org/10.1038/s41467-023-40112-4; Zeder et al. 2023, The effect of a short observational record on the statistics of temperature extremes, Geophysical Research Letters, https://doi.org/10.1029/2023GL104090). It requires many hundreds of projections to produce an extreme outside of the regular climatology. This research is yet to provide guidelines and datasets for the practitioner. We will note in the discussion of uncertainty that there could be events outside the boundaries of our simulation.

**Minor comments**

Use superscripts for units such as km2 and m3/s

- Thank you for noticing. We will make that change.

The discussion talks quite a bit about risk (Section 4.3). However, the authors do just look at changes in hazard while changes in vulnerability and exposure are not assessed. To avoid confusion, I would therefore use more specific terminology.

- Good point. We will outline in the text that our discussion of increasing or decreasing risk relates specifically to changing hazard.

---

## Author Comment (AC2)

**Author response to reviewer #2 comments for HESS manuscript "Modelling the effects of climate and landcover change on the hydrologic regime of a snowmelt-dominated montane catchment " [Paper #: hess-2023-248]**

Dear Reviewer,

We would like to thank you for taking the time to complete such a thorough and thoughtful review of the original manuscript submission. Addressing your comments will undoubtedly result in a stronger submission. Please find below a list of responses to your comments. We hope our responses satisfy the spirit and intent of your remarks.

Sincerely,

Russell Smith

**Reviewer #2 comments**

**General Comments**

Smith et al. applied a modelling approach to evaluate the combined effects of climate and landcover changes on the hydrologic regime of a snowmelt-dominated montane catchment. Based on the modelling results, management strategies that mitigate the negative impacts were identified. The study is regionally focussed on the Penticton Creek catchment, Canada, a catchment that was already in the focus of other studies before (e.g., Winkler et al., 2017; Winkler et al., 2021; Spittlehouse and Dymond, 2022; Smith, 2018, 2022). The authors address an important topic, in particular the interaction of the effects of forest fires (landcover) and climate change on hydrological regimes. Both will become increasingly important in the future.

Overall, the manuscript is well-written and provides insights on annual and seasonal hydrological changes under RCP 8.5 global warming and different landcover conditions. The percentage of disturbed area because of historical wildfires and forest harvesting was considered in landcover scenarios. In total one climate change scenario and 5 landcover change scenarios were parameterized and simulated with the Raven model. Figures and tables clearly show the results of various analyses.

The study is scientifically sound and based on methods that are well established in the scientific community. Although very case specific, the model results are useful for the development of water management strategies.

- Thank you for the positive comments.

**Individual scientific issues**

Line 46-47: Are findings for the Penticton Creek comparable? Results from Westra et al. (2013) are not discussed further.

- Comparing the highest daily precipitation in the records, the rate of increase is approximately 2.9% K$^{-1}$ from the current climate to the 2080s climate. This change is lower than the values reported by Westra et al. (2013). We will consider incorporating this point in the discussion of uncertainty when revising the manuscript.

Line 129: please add classification system for lower, middle, and higher elevations (in meter).

- Specific elevations are referenced in lines 345-346 for the more detailed snowpack sensitivity analysis; however, the reference to elevation at line 129 is for the purpose of describing the historical hydrologic regime, in general. We propose revising line 129 to the following: "A snowpack persists in upper areas of the catchment from October-November through April-June, and is intermittent in lower areas.

Table 1: was data from Penticton Airport met. station used in the study, if so, how was it considered?

- The model was calibrated on mean precipitation at Penticton Airport to constrain the precipitation lapse rate (see line 264). Observed air temperature lapse rates between P1 and Penticton Airport were used to define calibration constraints for air temperature lapse rates. These points will be clarified in the text.

Line 141-142: How were data spatially distributed?

- Precipitation and air temperature were distributed using calibrated lapse rates. This point will be clarified in the text.

Line 143: What algorithms were used (degree-day method?)

- The snowpack balance incorporated coupled mass and energy balance equations. The full snowpack energy balance was represented using algorithms that estimate energy fluxes using daily precipitation, and daily minimum and maximum air temperature (Quick 1995). It accounted for cloud cover, short-wave radiation, long-wave radiation, and turbulent flux (Quick, 1995; Dingman, 2002). These details will be provided in the supplementary.
- Quick, M., 1995. Computer models of watershed hydrology. Water Resources Publications, Highlands Ranch, Colorado. chapter The UBC Watershed Model. pp. 233–280.
- Dingman, S., 2002. Physical Hydrology. Waveland Press Inc.

Line 153: How was Greyback Lake considered in the model. This is not clear from the text.

- Greyback Lake is a controlled reservoir (p.4, l.100). Storage in Greyback Lake was "naturalized" so-to-speak. Bathymetric data and lake level data were combined to generate a time series of daily storage change. Increasing storage was added to the discharge for the catchment outlet (and vice versa for decreasing storage), assuming an instantaneous transfer to the catchment outlet. We deem this assumption reasonable with running the model at a daily time-step, as the actual transit time during high flow periods would be ~1 hour, and the rate of storage change during low flow periods would be very low. These details will be clarified in the text, and incorporated in the discussion of uncertainty.

Section 2.2.2:

Historical weather data (T, P) from one station located in the upstream area were used for model parameterization. Does the P1 station in the upper part of the catchment dominate the hydrological regime downstream? Should be discussed in uncertainty section.

- The model was run using data from the P1 weather station, but air temperature and precipitation at the Penticton Airport station were used to constrain lapse rates (see response above regarding Table 1). These points will be clarified in the text.

One climate change scenario (RCP 8.5) from one GCM (CSIRO) was considered in this study. The selection and (benefit?) of just one climate change scenario is discussed in the uncertainty section but is not based on other scientific results. Different patterns from other GCMs and timing (GCM and RCP dependent) might lead to different results. Here, the authors should explain in more detail why this scenario was selected and the benefit for this study.

- We acknowledge that simulating multiple climate scenarios is frequently used for projecting climate change impacts. Because of budget limitations, we had to choose between complexity of climate scenarios, land cover conditions, and hydrologic indicators. We decided to limit the climate scenarios by choosing one that had a severe climate change. Our rationale is that the scenario we chose would indicate how much hydrology may change and, thus, pose the greatest challenge to management. In doing so, we retained complexity in land cover because it's something forest and land managers can influence. We retained complexity in the hydrological indicators because of their importance to human values. However, we plan to address your concern by running four additional climate scenarios. We will provide additional tabular and/or graphical outputs in the manuscript as a sensitivity analysis, and incorporate these results in the discussion.

However, synthetic time series of 100 years on air temperature (min, max) and precipitation were prepared to simulate 100 years of river discharge and SWE. How close are the synthetic time periods to the GCM output? OR vice versa, how strong are baseline and future GCM time series biased to (fixed by) observations?

- We ensured that the synthetic climate change time series had similar statistical distributions of temperature and precipitation as the GCMs (see lines 190-194 in the manuscript; described in Spittlehouse & Dymond, 2022).

Regarding climate change, why were future GCM data on solar radiation and wind speed (wind direction) not used although these are key drivers of ET and snow storage processes?

- There is no solar or wind speed data in the GCM output that is available to us. The Raven model simulates short-wave radiation and turbulent flux using the approaches described above (see response above regarding Line 143). It is a standard procedure in many hydrological models (e.g., Tsuruta & Schnorbus 2021, Exploring the operational impacts of climate change and glacier loss in the upper Columbia River Basin, Canada, Hydrological Processes, https://doi.org/10.1002/hyp.14253).

Section 2.2.3:

Landcover scenarios seem to be static for a given year or vegetation status – is this correct? A 100-year time series of climate change input was combined with one-year landcover condition. If this is assumed, then it should be clearly stated in the text. [unfortunately, the reference Spittlehouse and Dymond is not freely accessible]

- The landcover conditions were static (they did not change over a 100 year simulation). This point is made in lines 91-92, and in lines 336-337. While each of the three synthetic weather records (current, 2050s, and 2080s) was run through the model as a 100-year time series, each 100-year record represented a stationary climate (see lines 179-180). In essence, each year in the 100-year record represented a different variation of a given climate. This approach allowed calculation of probability distributions from the hydrological outputs. These points will be clarified further in the text.

Line 200: What method was used for the interpolation? Please add!

- Using SAGA GIS, species was gap-filled using the nearest neighbour function, and density was gap-filled using the multilevel B-spline function. These points will be clarified in the text.

Line 221ff: Sorry, I've some difficulties in understanding how Fig. 3 was generated, particularly 3b and 3 c. Fig. 3b and c look different compared to the historical wildfires (Fig. 2g). There were wildfires in the outlet region in the past, but not visible in Fig. 3c?

- The disturbance distributions in Fig. 3b and 3c were based on simulated wildfires from LANDIS being imposed on the forested condition. The disturbance distributions in Fig. 3d and 3e were based on the natural disturbance history; thus, they align with Fig. 2g. These points are discussed in lines 195-199 and 221-232, but will be clarified further in the text.

Section 2.3.4:

Line 252-253: any reference that supports this suggestion? How is net negative snowfall interception considered in the model?

- This phenomenon (higher snowpack in low density forests than in nearby clearings) has been observed by the lead authour through snowpack monitoring in several high elevation forests within the southern interior of British Columbia (unpublished data) (e.g., Smith 2018, 2022). We're not aware of any peer reviewed literature that describes this pattern. However, for upper elevation stands, calibrated interception was allowed to be very small (e.g., 2%) in the model, but not negative. The point about net negative snowfall interception will be removed from the text, as it's unnecessary.

Line 259: What other model parameters were based on empirical observations? Please add examples to the manuscript or further descriptions to the SI.

- Good question. The text will be revised to indicate that other model parameters were assigned or constrained based on empirical observations. The following examples will be provided in the relevant sub-sections (an additional sub-section may be added for soils and stream channels):
  - Solar radiation from P1 station was used to assign the parameter value for cloud penetration.
  - Solar radiation and air temperature from P1 station were used to constrain parameters controlling cloudiness.
  - Air temperature and precipitation from P1 and Penticton airport stations were used to constrain lapse rates.
  - Air temperature and wind speed from P1 station were used to constrain parameters controlling atmospheric stability.
  - Measured snowpack albedo from Upper Penticton Creek was used to constrain albedo decay.
  - Soil mapping was used to assign soil porosity and texture, and to constrain soil depth and rates of percolation, interflow, and baseflow.
  - Leaf area index (LAI) from hemispherical photos was used to constrain LAI.
  - Manual snowpack observations were used to constrain throughfall (comparing forests and clearings), and to constrain snowpack patchiness during melt.
  - Satellite imagery was used to assign channel widths for the Penticton Creek mainstem channel.
  - Visual observations of the Penticton Creek mainstem channel were used to constrain channel roughness.

Section 2.3.1:

This sub-section needs more explanation. How were empirical relations developed and for which time period? Although this might be described somewhere else, more details are desirable here.

- See response immediately above. Relevant details will be provided in Section 2.3.1.

Section 2.3.2:

Line 295: ESSFdc1 = ESSFL? Please correct (in SI as well)

- You've identified an inconsistency in that line 250 states ESSFdc1 is "hereafter referred to as ESSFL"; however, we're inclined to list the actual BEC variant at line 295 to avoid confusion, as multiple other variants are also listed at line 295.

Line 301: this sentence is interesting. I would expect that throughfall increases with increasing precipitation (amount and duration), since more water drains towards the floor. In terms of forest hydrology, wind speed and wind direction are also key drivers of throughfall and its spatial variability. Please explain.

- Thank you for noticing this typo. The text will be revised to state that "throughfall percentage is expected to increase with increasing precipitation".

Section 2.3.3:

Which parameters were calibrated/optimized and how? Single parameters or multiple parameters at the same time? Are these parameters sensitive in variation?

- See response above regarding line 259. The corresponding details will be provided in the relevant sub-sections.
- Line 264 states that parameters were calibrated simultaneously. This point will be moved to Section 2.3.3.

Line 336: Refer to section on climate change scenarios.

- Will do. Thank you.

Line 360: Environmental risk is also impacted by shifts in seasonality.

- Agreed. Lines 359-362 are intended to outline how the event frequency analyses were defined. That is, higher yield is treated as higher risk for the peak flow frequency analysis, whereas lower yield is treated as higher risk for the low flow and annual discharge frequency analyses. These points will be clarified in the text.

Section 3.2:

Table 3: This table shows the mean rainfall (mm). For current conditions, e.g., 10 mm in winter = 10 mm over 3 months period excluding snow? Is the number/unit correct? Differs a lot from numbers for winter as given in Table 4 (although net precip., numbers are much higher)

- The values are higher in Table 4 because they include both snow and rain (minus ET), whereas Table 3 includes only rain. Under the current climate, almost all winter precipitation falls as snow at the high elevation.

Future numbers need to be compared to the baseline. Please add baseline numbers. Incremental change means absolute change?

- It is unclear whether you're referring to Table 3 or 4. By "baseline', we infer that you're meaning values for the current climate, which are provided in both tables.
- Yes, incremental change means absolute change. This point will be clarified in the text.

Section 3.4:

Regarding peak flow and summer low flow conditions, how were the highest and lowest flow events identified (Figs. 10 and 11)? Do the figures represent a graph of a specific year in which the highest or lowest event took place? Or were the highest or lowest values per month or day selected? (Same for lowest annual discharge, Fig. 12) The authors may think about

- For each climate condition and landcover, the selected peak flow event corresponds to the highest discharge in the simulated record. The discharge plotted in Figure 10 corresponds to the simulated flows for the actual event (i.e., they are event hydrographs).
- The same general approach was followed for Figures 11 and 12 (i.e., lowest discharge in the simulated record; event hydrographs), except that the selected events were the lowest 30-day mean discharge during the summer period and the lowest annual discharge, respectively. The summer period was defined as day of year 172 through 264.
- You may have noticed that Figure 11d shows lower flows in the October and February-March periods, but they are not within the summer period and, therefore, were not the basis for selecting the events.
- All of these points will be clarified in the text.

The authors should consider not splitting chapter 3.4 into sub-chapters, as each sub-chapter begins with the same wording and is structured in the same way. The results can be presented more concisely,

- Thank you for the suggestion. We considered combining those sections, as there is repetition in the overall approach to presentation. However, almost all of the actual content is unique to each section. It's our preference to keep the sections separate, to maintain clarity in the discussion of event type and associated hydrometeorological conditions.

Figure 12 can be moved to the SI.

- We infer that this suggestion came from Figures 11 and 12 looking similar; however, they represent different years in the synthetic weather records (for each climate condition, Figure 11 is the year with lowest summer low flow, and Figure 12 is the year with lowest annual discharge). Moreover, there are some important distinctions between the two figures that are relied upon. In particular, Figure 11a shows generally higher net precipitation in the winter and early spring periods for the 2050s and 2080s climates, compared to Figure 12a, but much lower net precipitation in May and somewhat lower in June. Figure 11c shows snowpacks that are normal or somewhat low but melt early, whereas Figure 12c shows snowpacks that are well below normal. These distinctions contribute to

inferences that the volume of snowpack and spring freshet runoff are important controls on annual discharge, whereas the timing of the spring freshet recession flow is more important than the volume of spring freshet runoff in controlling the severity of summer low flow. For these reasons, we prefer to retain Figure 12 in the main body; however, we will clarify these distinctions and inferences in the text, and also by presenting Figures 11 and 12 in a side-by-side arrangement.

Section 4.1:

Line 667: What is meant by rainstorm size?

- Thank you for inquiring about this point. We made a typo, as Lewis et al referred to storm size with respect to peak discharge, not rainfall amount. We propose revising our text to the following: "In this respect, Lewis et al. (2001) found that logging induced relative increases in peak discharge declined with increasing event magnitude".

Section 4.2:

Line 705: What does the text "This mitigative influence …" refer to? Are the large burn condition the mitigative influence?

- Yes. This point will be clarified in the text.

Line 730: Summer net precipitation is negative and will further decrease. Where does the water come from that drives an increasing ET?

- Good question. Summer net precipitation at 1800 m (Table 4) decreases substantially under the 2050s climate, then generally increases slightly under the 2080s climate. At this same elevation, snowpack melt-out occurs in May and June under the current climate, May under the 2050s climate, and April and May under the 2080s climate (Fig. 4d). We infer that soil moisture retention from snowmelt and spring rainfall is the primary water source for summer ET under the current and 2050s climates. We also infer that summer ET becomes supply limited under the 2080s climate. We will incorporate these points in the text.

Section 4.3:

Hydrological risk – I am not sure if "risk" is the best word to be used here. Risk means information on the exposure and vulnerability. A probability of the occurrence of floods and low flows is provided, however, no information is provided on e.g. damages or losses.

- Good point. We will outline in the text that our discussion of increasing or decreasing risk relates specifically to changing hazard.

Line 785: How certain are scenario runs for the 2050s?

- See response immediately below.

Section 4.4

Line 786-794: Input from one GCM and one RCP was used to drive the hydrological model. Best guesses on a lower RCP 4.5 are described under this section without any references or model results. It is most likely that RCP 8.5 leads to higher peak flows and lower summer discharges compared to RCP 4.5 or RCP 2.6. Of course, the argumentation holds true for results expected from lower emission scenarios, but different GCMs may lead to different precipitation patterns and intensities. In terms of identifying management strategies, it is questionable if "a worst-case scenario" is the best choice. I believe it would be advisable to draw conclusions for (sustainable) management from model results driven by the input of a climate ensemble to identify a robust solution.

- See response above related to simulating additional climate scenarios.
- With respect to modelling a "worst case scenario", based on the current global trajectory of emissions related decision making, it seems unlikely that lower emissions scenarios will play out, at least with respect to the 2050s climate (if not the 2080s climate). We deemed it more prudent to model a higher emissions scenario, to identify and communicate key risks to society and opportunities for mitigation.